# Random Noise Attenuation of Sparker Seismic Oceanography Data with Machine Learning

Hyunggu Jun [1], Hyeong-Tae Jou[1], Chung-Ho Kim[1], Sang Hoon Lee[1], Han-Joon Kim[1]

[1]Korea Institute of Ocean Science & Technology, Busan, 49111, Republic of Korea

*Correspondence to*: Hyeong-Tae Jou (htjou@kiost.ac.kr)

**Abstract.** Seismic oceanography (SO) acquires water column reflections using controlled source seismology and provides high lateral resolution that enables the tracking of the thermohaline structure of the oceans. Most SO studies obtain data using air guns, which can produce acoustic energy below 100 Hz bandwidth, with vertical resolution of approximately ten meters or more. For higher-frequency bands, with vertical resolution ranging from several centimeters to several meters, a smaller, low-

cost seismic exploration system may be used, such as a sparker source with central frequencies of 250 Hz or higher. However, the sparker source has a relatively low energy compared to air guns and consequently produces data with a lower signal-to-noise (S/N) ratio. To attenuate the random noise and extract reliable signal from the low S/N ratio of sparker SO datawithout distorting the true shape and amplitude of water column reflections, we applied machine learning. Specifically, we used a denoising convolutional neural network (DnCNN) that efficiently suppresses random noise in a natural image. One of the most

important factors of machine learning is the generation of an appropriate training dataset. We generated two different training datasets using synthetic and field data. Models trained with the different training datasets were applied to the test data, and the denoised results were quantitatively compared. To demonstrate the technique, the trained models were applied to an SO sparker seismic dataset acquired in the East Sea and the denoised seismic sections were evaluated. The results show that machine learning can successfully attenuate the random noise in sparker water column seismic reflection data.

# 1 Introduction

Conventional physical oceanography measurements from cruises are performed by deploying instrument such as a conductivity-temperature-depth (CTD), an expendable conductivity-temperature-depth (XCTD) or an expendable

bathythermograph (XBT) at observation stations. In general, due to time and cost limitations, the distance between observation points is large, from hundreds of meters to tens of kilometers; thus, the acquired water column information has a low horizontal resolution. Seismic oceanography (SO) is a method that obtains the water column reflections via seismic exploration and analyzes seismic sections to estimate the oceanographic characteristics of sea water. This method was first implemented by Gonella and Michon (1988) to measure deep internal waves in the eastern Atlantic and later became popularized after the work

of Holbrook et al. (2003). Spatial differences in temperature and salinity generate an acoustic impedance contrast that results in the reflection of seismic signals. The reflected seismic signals are processed to image the thermohaline fine structure of the ocean (Ruddick et al., 2009). Seismic exploration acquires data continuously in the horizontal direction; thus, it has the advantage of generating data with improved horizontal resolution relative to that obtained by conventional probe-based oceanographic methods (Dagnino et al., 2016). Therefore, SO is used to image the structure of water layers (Tsuji et al., 2005;

Sheen et al., 2012; Piété et al., 2013; Moon et al., 2017) and provide quantitative information, such as the physical properties (i.e, temperature, salinity) (Papenberg et al., 2010; Blacic et al., 2016; Dagnino et al. 2016; Jun et al., 2019) or the spectral distribution of the internal waves and turbulence (Sheen et al., 2009; Holbrook et al., 2013; Fortin et al. 2016) after analysis where temperature or salinity contrasts produce clear seismic reflections.

SO has been conducted mainly using air guns, a high-energy source, and the central frequency, the geometric center of the

frequency band (Wang, 2015), of air guns is usually below 100 Hz. Therefore, the vertical resolution of the acquired seismic data using air guns is approximately ten meters or more, which is coarser than that of conventional physical oceanography observation equipment. SO also has the disadvantage of high exploration expenses when using air guns and streamers several kilometers long, which require large vessel and many operators. Ruddick (2018) highlighted the limitations of current SO studies using multichannel seismic (MCS) exploration and argued that using a small-scale source instead of a large-scale air

gun and a relatively shorter streamer with a length shorter than 500 m can make SO more widely available.

Piété et al. (2013) implemented a sparker source with a central frequency of 250 Hz and a short 450-m streamer (72 channels at 6.25 m intervals) to examine the oceanographic structure. Since high-frequency band sources were implemented, data with vertical resolution theoretically as fine as 1.5 m were acquired, and the short source signature enabled the thermocline structure to be imaged even in very shallow areas between 10 and 40 m. However, the signal-to-noise (S/N) ratio of the seismic section

was lower than that of the air gun source, and the amplitude of the thermocline feature was small; thus, it was difficult to interpret. Generally, using a low-energy source and a short streamer in seismic exploration causes the low-S/N ratio problem. This problem is more accentuated in SO because the maximum impedance contrasts between the water layers are smaller than the maximum impedance contrasts between the layers beneath the seabed. If a low-energy source is used, the water column reflections recorded by the receiver become too weak, and the influence of the background noise becomes larger than when

using a high-energy source. The improvement in vertical resolution is evident when using higher-frequency band sources such as a sparker source; therefore, if appropriate methods can effectively suppress the random noise in the seismic section, more useful information can be derived compared to SO data using an air gun source.

There are various types of noise recorded by the receiver in seismic exploration, and several data processing steps are usually applied to the seismic data to attenuate noise. However, the noise attenuation method not only removes noise but also potentially alters important seismic signals (Jun et al., 2014). Especially for SO data, careful processing is essential to recover the actual shape of the water column reflections (Fortin et al., 2016), which contain internal wave and turbulence information. It is difficult to apply various noise attenuation methods to SO data because analyzing the internal wave and turbulent subranges of the water column requires the horizontal wavenumber spectrum (Klymak and Moum, 2007) of the seismic data, which is liable to be damaged by data processing. Therefore, minimized noise attenuation processes have been applied to SO data, and for this reason, studies calculating the wavenumber spectrum by using SO data such as those by Holbrook et al. (2013) and Fortin et al. (2016, 2017) have only applied bandpass and notch filters to remove random and harmonic noise. However, when the sparker is used as a seismic source, the bandpass filter alone is not sufficient to attenuate random noise, resulting in great difficulties in analyzing the wavenumber spectrum. Therefore, it is necessary to apply additional data processing to properly attenuate noise without damaging the wavenumber characteristics of SO data.

The use of artificial intelligence (AI) has been studied in geophysics for decades (McCormack, 1991; McCormack et al., 1993; Van der Baan and Jutten, 2000), but recent advances in computer resources and algorithms have spurred AI research, and several studies have been conducted to apply machine learning in the field of seismic data processing (Araya-Polo et al., 2019; Yang and Ma, 2019; Zhao et al., 2019). Among them, one of the most actively studied areas is prestack and poststack data noise attenuation. After convolutional neural networks (CNNs) were introduced, various noise attenuation methods based on the CNN architecture have been proposed (Jian and Seung, 2009; Gondara, 2016; Lefkimmiatis, 2017), and the denoising convolutional neural network (DnCNN) suggested by Zhang et al. (2017) attained good results in random noise suppression in natural images. Recently, the DnCNN was applied to attenuate various types of noise in seismic data (Li et al., 2018; Si and Yuan, 2018; Liu et al., 2020). The DnCNN uses residual learning (He et al., 2016) and has the advantage of minimizing damage to the seismic signal by estimating the noise from seismic data rather than directly analyzing the signal. The original shape of the water column reflector in SO data remains unchanged during data processing, so the DnCNN, which learns noise characteristics, is a suitable SO data denoising algorithm.

As important as the proper neural network architecture when conducting training through machine learning is the use of an appropriate training dataset. When using the DnCNN to attenuate noise, the training data require noise-free and noise-only (or noise containing) data. In this study, we use both field and synthetic data as training data and compare which training data are more suitable for the DnCNN in attenuating random noise in SO data.

First, we introduce the DnCNN architecture used in this study and explain the construction method for the training and test datasets using field and synthetic data, respectively. Then, we perform training using the constructed training datasets and

verify the trained models using test datasets. Finally, the trained models are applied to the East Sea sparker SO data, and the results are compared and evaluated.

90

## 2 Data and Methodology

### 2.1 Review of the DnCNN

The purpose of this study is to attenuate the random noise in sparker SO data, and the machine learning architecture used in this study is the DnCNN, which was suggested by Zhang et al. (2017). DnCNN is a neural network architecture based on the CNN for the purpose of removing the random noise in natural images. DnCNN reads the noisy image in the input layer and extracts the noise from the noisy image in the hidden layer. A layer is a module containing several computing processes (e.g. convolution, pooling or activation). At the output layer, the extracted noise is subtracted from the noisy image and generates the denoised result. The DnCNN has three distinctive characteristics: 1) residual learning, 2) batch normalization, and 3) the same input and output data size for each layer.

Residual learning was first suggested by He et al. (2016) and the neural network which uses residual learning (residual network) contains several residual blocks. The residual block is a building block of the residual network and consists of several convolution processes and a shortcut connection. The DnCNN adopted residual learning and used a single shortcut to estimate the noise from natural images. The estimated noise was subtracted from the noisy natural image, and the noise-attenuated image remained. If the DnCNN is applied to seismic data denoising, the target noise is estimated from the noisy prestack or poststack seismic data, and the estimated noise is subtracted from the noisy seismic data. The seismic data including noise ($y$) can be expressed as the sum of the true seismic data ($x$) and noise ($n$) as follows:

$$y = x + n. \tag{1}$$

When the deep learning architecture that estimates noise from the noisy seismic data is $\boldsymbol{D}(y; n)$, the cost function of the DnCNN (C) can be expressed as follows:

$$C = \frac{1}{2N} \sum_{i=1}^{N} \|\boldsymbol{D}(y_i; n_i) - (y_i - x_i)\|^2, \tag{2}$$

where $n$ is the estimated noise from the original noisy seismic data ($y$), N is the number of the training data and $\| \quad \|^2$ is the sum of squared errors (SSE). Although the DnCNN uses residual learning, it is different from the conventional residual network. The conventional residual network utilizes residual learning to solve the performance degradation problem when the network depth increases; thus, it includes many residual blocks. On the other hand, the DnCNN uses residual learning to predict noise from noisy images and includes a single residual block. For example, ResNet (He et al., 2016), which is a well-known image recognition network using residual learning, has more than tens or hundreds of network depth layers with many residual blocks, but the DnCNN has fewer than 20 network depth layers with a single residual block. Moreover, the DnCNN applies batch normalization (Ioffe and Szegedy, 2015) after each convolution layer. Instead of using the entire training data at the same time, the machine learning sequentially uses mini-batches that represent a small part of the entire data as input for efficient training.

The distribution of input data varies during training and the neural network has a risk of updating the weights in the wrong direction. Batch normalization is a method to normalize the distribution of each mini-batch by making the mean and variance of the mini-batch equal to 0 and 1, respectively. The normalized mini-batch is transformed through scaling and shifting. Batch normalization is widely used in many deep learning neural networks because it can stabilize learning and increase the learning speed (Ioffe and Szegedy, 2015). The authors of the DnCNN empirically found that residual learning and batch normalization

create a synergistic effect. In addition, unlike the encoder-decoder type denoising architecture, the size of the input data of the DnCNN is the same as the size of the output data in each layer. The DnCNN directly pads zeros at the boundaries during convolution and does not contain any pooling layer; thus, the data size remains unchanged during training. This procedure has the advantage of minimizing the data loss occurring during the encoding and decoding process. As mentioned above, the amplitude and shape of the seismic reflections are important for spectrum analysis using SO data. To minimize possible

deformation of the seismic signals during the denoising procedure, the DnCNN, which predicts noise using residual learning and avoids information loss due to the absence of an encoding-decoding model, could be an appropriate algorithm.

## 2.2 Network architecture

The DnCNN uses three different kinds of layers, and we use the same layers as suggested by Zhang et al. (2017). Fig. 1 shows

the DnCNN architecture used in this study, where Conv, BN, and ReLU indicate convolution, batch normalization, and rectified linear units (Krizhevsky et al., 2012), respectively.

The first layer type consists of convolution + rectified linear units and is used only at the first layer of the network architecture. This layer is shown as "Conv+ReLU" in Fig. 1. In the convolution process, 2-dimensional convolution between a certain size of kernel and data is performed. The outputs of the convolution process are passed through the ReLU activation function which

returns input value for the positive input and 0 for the negative input to add the nonlinearity in the network (Huang and Babri, 1998). The size of the convolution filter is $3 \times 3 \times c$ and generates 64 feature maps which is the same number of feature maps in Zhang et al. (2017), where $c$ is the number of channels of the input data. This study extracts noise from binary files, thus a $3 \times 3 \times 1$ convolution filter is adopted. The second layer type consists of convolution + batch normalization + rectified linear units and is applied from layers 2 to L-1, where L is the total number of network layers. This layer is shown as

"Conv+BN+ReLU" in Fig. 1. Sixty-four $3 \times 3 \times 64$ convolution filters are used because the number of feature maps of the hidden layer is 64, which is the same for all hidden layers. After convolution, batch normalization and the ReLU activation function are applied. The third layer type is convolution and uses only the last layer to generate output noise data, and one $3 \times 3 \times 64$ convolution filter is used. This layer is shown as "Conv" in Fig. 1. After training is completed, the predicted noise is subtracted from the input data to produce denoised data.


**2.3 East Sea SO data**

The purpose of this study is to attenuate the random noise in the East Sea sparker SO data. The East Sea sparker SO data were obtained with a 5,000-J SIG PULSE L5 sparker source to investigate the propagation of the internal tide and characteristics of turbulent mixing. Two seismic lines were explored: line 1 traveled from southwest to northeast, and line 2 traveled from

northeast to southwest (Fig. 2). The survey was performed from October 7th to 11th in 2018 (approximately 38 hours for one line) and the vessel speed was 5.5 knots. The seismic data include the shallow continental shelf and slope with a water depth of ~ 200 m, but we removed the continental shelf and slope area and used 280.4 km of line 1 and 280.9 km of line 2 because the data from these sections did not target the layers below the sea floor but the water layer. The shot interval was approximately 15 m, and 24 receivers were used at intervals of 6.25 m.

The acquired seismic data were processed through conventional time processing consisting of instrument delay and amplitude corrections, bandpass filtering, common-midpoint (CMP) sorting and stacking. Amplitude correction was performed by empirically multiplying by the square root of time at each time step. The corner frequencies of the trapezoidal bandpass filter (Dickinson et al. 2017) were 60-80-250-300 Hz, which were higher than those in air gun seismic data processing. Sparker source data have a lower S/N ratio due to the weak energy source compared to air gun source data and generally rely on a

shorter streamer length; thus, it is common to generate supergathers (Piété et al., 2013) to enhance the S/N ratio. We combined 4 neighboring CMP gathers (Tang et al., 2016) to construct one supergather. A constant sound speed of 1500 m s$^{-1}$ was adopted for normal move-out. After CMP stacking, data recorded before 0.03 s were eliminated from the stack section because only direct waves and noise were present, and water layer reflections were rarely recorded. The processed seismic sections are shown in Fig. 3. The calculated vertical and horizontal resolution (Yilmaz, 2001) of the processed seismic section are

approximately 1.5 m and 17.3 m, respectively, when the central frequency is 250 Hz, sound speed is 1500 m s$^{-1}$, and reflector depth is 100 m. The internal waves in the research area propagate above a depth of 200 m, which is approximately 0.26 s in the seismic section. In addition, the physical properties of the research area were measured with oceanographic equipment, such as XCTD and XBT, during exploration. Fig. 4 (a) shows the temperature profiles from two XBTs and two XCTDs. From the measurement data, the mixed layer ranged from the sea surface to a depth of 30 m, the depth of the thermocline ranged

approximately from 30 to 200 m and deep water occurred below approximately 200 m depth. Fig. 4 (b) shows the reflection coefficients, defining the ratio between the reflected and incident wave, calculated with the XBT (assuming a constant density 1 g cm$^{-3}$) and XCTD data. The reflection coefficients are very small (~0.00005) at depths shallower than 30 m, which seems to be the mixed layer, and deeper than approximately 200 m, which seems to be the deep water layer. Deep water exhibits a very slight water temperature/salinity variation with the depth, which makes it difficult to generate reflections, as indicated by

the seismic sections and reflection coefficients. Therefore, data after 0.28 s are considered random noise, and we used this part as noise data for the DnCNN. Random noise in the seismic data can be created by rough weather conditions, ocean swells, tail buoy jerks, the engine and propeller of the vessel, etc. For conventional noise attenuation methods, it is important to estimate

the noise sources and their properties. In contrast, for machine learning-based noise attenuation suggested in this study, these properties do not need to be estimated because noise itself is used as training data.


## 2.4 Training data

The most important noise attenuation aspect of machine learning is generating an appropriate training dataset. Noise-free seismic sections (the ground truth) and sections with noise are required to generate the training dataset, and the training dataset can be constructed by combining these two datasets. As previously explained, the purpose of this study is to effectively

attenuate noise in the water column seismic section acquired in the East Sea. Thus, the noisy section can be easily obtained by extracting the deep water zone of the water column seismic section without reflections. At this point, we assume that the random noise of the top and bottom parts of the water column seismic section exhibits similar features. The noise parts of the East Sea SO data are shown as red boxes in Fig 3. There are no notable reflections in the noise parts. However, it is almost impossible to obtain noise-free seismic sections from field data. Therefore, we constructed training datasets using two different

methods and compared these datasets.

Training dataset 1 obtains the ground truth based on the field sparker seismic section below the sea floor. The reflection coefficients of the major reflectors below the sea floor are tens to hundreds of times larger than those of the water column; thus, the seismic data below the sea floor have a better S/N ratio than the SO data. In addition, after the proper data processing steps, the S/N ratio of the seismic data beneath the sea bed can be further enhanced. We used 14 lines of field sparker seismic

data targeting below the sea floor (SEZ data) acquired with the same equipment used to record the East Sea SO data. We used the interval from 0.2 to 0.6 s of the original data where the noise level is lower than in other parts of the data. A bandpass filter, FX-deconvolution, a Gaussian filter and noise muting above the sea floor were applied. Fig. 5 (a) shows an example of the SEZ data used to generate training dataset 1. This method has the advantage of using data with similar characteristics to those of the target data (the East Sea SO data) as the ground truth because the data are collected by the same equipment. Even if the

S/N ratio of the sparker seismic data beneath the sea bed is relatively higher than that of the sparker seismic data of the water column and noise is suppressed during processing, it is difficult to completely eliminate noise from seismic data. Therefore, this method has the disadvantage that there is a possibility that the remaining noise would have a detrimental effect on training. Training dataset 2 uses synthetic data as the ground truth. The method for generating a synthetic seismic section from the velocity model is to perform time or depth domain processing using prestack synthetic data or to convolve the reflection

coefficient and source wavelet. The former method has the advantage of generating synthetic seismic sections with features more similar to those of the actual field seismic section, but the generation and processing of prestack data are time consuming, and artificial noise is often generated during processing. The latter method has the advantage of generating noise-free seismic sections with a very simple procedure. However, the generated synthetic seismic section has much different features from the target seismic section, which is, in this study, the East Sea water column sparker seismic section. Therefore, when the trained

model is applied to the target seismic section, there is a risk that the trained model will regard the reflection signal as noise. In

this study, we used the latter method to generate the ground truth because we needed to avoid artificial noise. Marmousi2 (Martin et al. 2006) and Sigsbee2A (Den Bok, 2002) synthetic velocity models with a constant density ($1\,\text{g}\,\text{cm}^{-3}$) were employed to calculate the reflection coefficient, and the first derivative Gaussian wavelet was the synthetic source wavelet. The original Marmousi-2 and Sigsbee 2A synthetic velocity models are depth domain velocity models, but we assumed that these velocity models were time domain models to generate time domain seismic sections via 1-dimensional convolutional modeling. Fig. 5 (b) and (c) show the generated seismic sections of Marmousi-2 and Sigsbee 2A, respectively.

Each ground truth was first divided into $300 \times 300$ sections. Then, amplitude values higher than the top 1% and lower than the bottom 1% were replaced by the top 1% and bottom 1% values, respectively, to prevent outliers from significantly affecting training. In addition, the outlier-removed ground truth and noise section were normalized to the maximum value of each section. This procedure balances the amplitudes of the ground truth and noise before generating the training dataset. Finally, training data with field seismic noise were generated by combining the ground truth and noise at a random ratio. Eq. 3 is the method to construct the training data, and Fig. 6 shows an example of the training data compilation.

$$T = r_1 \times G + r_2 \times N, \qquad\qquad (3)$$

where $T$ is the noise-added seismic patch (training data), $G$ is the ground truth patch, $N$ is the noise patch extracted from the noisy part of the East Sea seismic section (noisy data), and $r_1$ and $r_2$ are random values ranging from 0.2~0.8 ($r_1 + r_2 = 1$). The dimensions of $T, G$ and $N$ are $50 \times 50$; $G$ and $N$ were extracted at a random location of the ground truth and noisy section. To increase the number of ground truth data, data augmentation was applied by zooming in/out and randomly rotating or flipping the data. The number of training data used in the training is determined by the size of the mini-batch and the number of iterations per epoch. Each training cycle using one mini-batch is an "iteration" and each training cycle using all the training data is an "epoch". If one mini-batch passes through the training, then one iteration ends. If all mini-batches pass through the training and all the training data has been used for training, then one epoch ends. In this study, training data were newly generated at every epoch with the fit_generator function in Keras (Keras Documentation, 2020). The mini-batch size was 128 and the number of iterations within an epoch was 220; thus, the fit_generator function generated 28,160 training data patches at every epoch.

### 3 Training

#### 3.1 Experimental setting

The experiment was conducted using 28,160 training data patches per epoch, and the size of each patch was $50 \times 50$. The mini-batch size was 128, the network depth which is the total number of layers in the network architecture was 17, the number of feature maps of each layer was 64 and the Adam optimizer (Kingma and Ba, 2015) was implemented by following Zhang et al.'s (2017) DnCNN experiments. The network architecture used in this study is shown in Fig. 1. We performed training by using the two different training datasets generated from the field data (training dataset 1) and synthetic data (training dataset 2). The DnCNN model was trained for 40 epochs, and the total training time was approximately 1 hour using a single NVIDIA Quadro P4000 GPU.

#### 3.2 Experiment using training dataset 1

Training dataset 1 was generated with the SEZ field data and noise obtained from the East Sea seismic section. After training the DnCNN model (D1 model) using training dataset 1, we evaluated the trained model against the test data. The test data were generated with the same procedure as that for the training data, and we used the other lines of SEZ data that were not used to generate the training data. Eighty-six $300 \times 300$ size test data were divided into $50 \times 50$ size patches, which is the same size as the training data patch, and 3,096 patches ($50 \times 50$) were generated. Among 3,096 patches, we used 3,072 patches which was 24 times the mini-batch size (128) for the test because of computational efficiency. Fig. 7 shows 6 randomly selected test data subset patches, ground truths and denoised results after applying the D1 model at the $5^{th}$, $10^{th}$, $20^{th}$ and $40^{th}$ epochs. The depicted test data patches (1 to 6) include noise, but most of the noise has been successfully removed after training for 40 epochs. In the $3^{rd}$ and $6^{th}$ patches of test data subset in particular, the reflections are hardly recognized because of the severe noise, but the D1 model successfully attenuated the noise and generated a denoised section almost identical to the ground truth. In addition, there is a water layer without any signal at the top of the $4^{th}$ test data patch, and the trained model properly attenuated the noise at the water layer. This means that the trained model can determine those parts where no signal occurs.

However, the trained model using training dataset 1 has one problem. The ground truth of the $5^{th}$ test data patch contains noise in the bottom right part, and training dataset 1 might also contain noise in some parts of the ground truth. Although the ground truth of training dataset 1 was generated from a processed sparker seismic section below the sea floor, noise still remained because it is almost impossible to perfectly remove noise from field data. The ground truth of training dataset 1 (SEZ data in Fig. 5 (a)) is obtained using the same equipment as was used for the East Sea SO data, which is the target of this study. Therefore, the ground truth signal has similar characteristics to the signal of the East Sea SO data, but its noise feature could also be similar to the noise of the East Sea SO data. This means that noise with similar characteristics would be trained to be eliminated in some cases and not in other cases during training. Training inconsistency can degrade the performance of the trained model.

To evaluate the test result quantitatively, we calculated the peak S/N ratio (PSNR) and structural similarity index measure (SSIM) by using entire test data. The PSNR reflects the amount of noise contained in the data and can be calculated as follows (Hore and Ziou, 2010):

$$PSNR = 20\,log_{10}(MAX_I) - 10\,log_{10}(MSE)$$

(4)

where $MAX_I$ is the maximum value of the image and MSE is the mean squared error between the data with and without noise. The PSNR is high when noise is successfully removed, while the PSNR is low when noise is not sufficiently removed. Fig. 8 (a) shows the average PSNR and standard deviation of the test results. At the early stage of training, the average PSNR is low, which indicates that noise has not been sufficiently removed, but it increases as training progresses and converges at approximately 36 dB after 25 epochs. Although the denoising algorithm attenuates noise successfully, the reflection shape, which is important information of the SO data, can be altered. Therefore, it is necessary to measure the structural distortion to verify the effectiveness of the proposed method. The SSIM is a quality metric that calculates the structural similarity between two datasets and can be calculated as follows (Hore and Ziou, 2010):

$$SSIM = \frac{(2\mu_x\mu_y + c_1)(2\sigma_{xy} + c_2)}{(\mu_x^2 + \mu_y^2 + c_1)(\sigma_x^2 + \sigma_y^2 + c_2)}$$

(5)

where $\mu$ is the average, $\sigma^2$ is the variance, $\sigma_{xy}$ is the covariance of the reference image ($x$) and test image ($y$), and $c$ is a stabilizing parameter. The value of SSIM ranges from 0 to 1, and if the structure is distorted during the denoising process, the SSIM will be low. On the other hand, the SSIM will be close to 1 if the denoised data are similar to the ground truth. Fig. 8 (b) shows the average SSIM of the test results. Similar to the PSNR result, the SSIM is also low at the early stage of training but increases as training progresses and converges at approximately 0.88 after 21 epochs. We also plotted the PSNR and SSIM histogram of the test data before applying the D1 model and after applying the D1 model (40[th] epoch) in Fig. 9. Both the PSNR and SSIM are clearly improved after applying the D1 model.

For seismic data, it is important to determine how well the actual amplitude and shape of the true reflection are recovered through the denoising process. Therefore, we extracted seismic traces from the denoised section and ground truth and compared the extracted traces, as shown in Fig. 10, to ensure that the trained model recovers the actual amplitude of the signal. We extracted the 20[th] (Fig. 10 (a)) and 30[th] (Fig. 10 (b)) vertical traces from the last (6[th]) patch of the test data, which had a size of 50x50. For the denoised trace, we extracted trace from the denoised patch of the 40[th] epoch. The amplitude and shape of the trace from the noisy data are different from those of the ground truth because the data are severely contaminated with noise. Although a considerable amount of noise is observed, the denoised traces have similar amplitudes and shapes to those of the ground truth. These results indicate that the DnCNN can recover important information of the true reflections and can be useful for random noise attenuation of sparker SO data.

### 3.3 Experiment using training dataset 2

Training dataset 2 was generated by using the modified Marmousi2 and Sigsbee2a synthetic seismic sections and noise obtained from the East Sea seismic section. After training the DnCNN model (D2 model) with training dataset 2, we evaluated the trained model against test data. The test data were generated with the same procedure used to generate the training data, and we selected part of the 1994 Amoco static test dataset (SEG Wiki, 2020), which is a different model from that used for the training data. The size of the test data patch was the same as that of the training data patch ($50\times50$), and the number of test

data patches was 3,072. Fig. 11 shows 6 randomly selected test data subset patches, ground truths and denoised results after applying the D2 model at the 5th, 10th, 20th and 40th epochs. Although the test data patches contain noise at different levels, the trained model at the 40th epoch attenuated most of the noise successfully and generated almost identical seismic sections to the ground truth. The second test data patch contained relatively little noise compared to other test data patches, and most of the noise was removed after approximately 10 training epochs. Test data patches 1 and 3 contained simple reflections with much

noise, and the noise was sufficiently removed after approximately 20 training epochs. The noise in test data patches 4 and 6 was more severe than the noise in the other test data patches. After 40 training epochs, most of the noise was attenuated but not perfectly removed. The noise was dominant in test data patch 5, and only a weak signal existed in the bottom part of ground truth 5. If we evaluate the denoised result of the 5th test data patch, noise had been successfully removed, and only a weak signal remained in the bottom part of the patch after 40 training epochs. This indicates that the trained DnCNN model can

accurately discriminate between signal and noise.

 Unlike training dataset 1, training dataset 2 was generated with synthetic data. Therefore, it has the advantage of using noise-free seismic sections as the ground truth. In addition, generating many different kinds of synthetic seismic sections does not require much time or effort; thus, it is easy to increase the amount of training data compared to using field data as training data. However, the features of synthetic seismic sections can be different from those of the target data requiring noise attenuation

because the synthetic seismic sections were generated by simply convolving the reflection coefficient with the source wavelet. Several studies have applied machine learning to field seismic data by training the model using synthetic data, such as automated fault detection with synthetic training data (Wu et al., 2019), but machine-learning-based noise attenuation of SO data using synthetic training data has not yet been studied.

 Similar to the first experiment, the average PSNR (Fig. 12 (a)) and SSIM (Fig. 12 (b)) converged after approximately 25

epochs. The histograms of PSNR and SSIM of the test data before applying the D2 model and after applying the D2 model (40th epoch) are also plotted in Fig. 13. As shown, the PSNR and SSIM are improved after DnCNN is applied. The average PSNR and SSIM in the second experiment are higher than those in the first experiment. These results could be caused by the use of a noise-free synthetic seismic section as the ground truth of training dataset 2 and might indicate that training dataset 2 is more appropriate for random noise attenuation of SO data. Fig. 14 shows the extracted traces before and after applying the

D2 model. We extracted the 20th (Fig. 14 (a)) and 30th (Fig. 14 (b)) vertical traces from the 1st patch of the test data. The

denoised traces successfully recovered the true amplitude and shape, although the input data were severely contaminated by random noise.

In the second experiment, the noise-attenuated traces are closer to the ground truth traces than those in the first experiment. However, the comparison of the several extracted traces does not indicate which training data are more suitable for suppressing noise of sparker SO data. Therefore, we calculated the root-mean-square (RMS) error between the denoised test data and ground truth of the test data and evaluated which training data produced a lower RMS error. The RMS error was calculated as follows:

$$RMS\ error = \sqrt{\frac{1}{ntest} \sum_{i=1}^{ntest} \sum_{j=1}^{nnode} \left(g_{ij} - d_{ij}\right)^2} \tag{6}$$

where $g$ is the ground truth of the test data, $d$ is the denoised test data, $ntest$ is the number of test data patches (3,072) and $nnode$ is the size of each data patch (50×50). Although test datasets 1 and 2 were generated using the same noisy data (the part containing noise of the East Sea SO section), the initial RMS errors of test datasets 1 and 2 before noise attenuation were different, 6.37 and 6.34, respectively, because noise was randomly extracted from the noise data. Therefore, we normalized the RMS error by that of the test data before noise attenuation. Fig. 15 illustrates the normalized RMS error of the first and second experiments at every epoch, and the normalized RMS errors were properly decreased in both results. The normalized errors converged at 0.27 in the first experiment and at 0.15 in the second experiment. The normalized RMS error of the second experiment is lower than that of the first experiment, indicating that the performance of the D2 model is better.

### 3.4 Calculation of the data slope spectrum from the synthetic seismic section

Water column reflection data can be used to obtain the physical oceanographic information by calculating the slope spectrum.
The data slope spectrum is a slope spectrum obtained directly from the seismic amplitude instead of tracked seismic reflections. The obtained horizontal wavenumber ($k_x$) spectrum of the seismic reflection amplitude is multiplied by $(2\pi k_x)^2$ to produce a data slope spectrum, which is useful for identifying noise contamination of seismic data to reveal the cutoffs from an internal wave to turbulence subrange (Holbrook et al., 2013; Fontin et al., 2017). Holbrook et al. (2013) suggested analyzing the data slope spectrum for the complete data before calculating the slope spectrum from the water reflections because the noise that
needs to be suppressed is more evident in the spectrum of the complete data. Therefore, we calculated and compared the data slope spectrum of noise-free, noise-added and noise-attenuated seismic data by using synthetic seismic section to verify that the proposed denoising method can recover the true data slope spectrum. The synthetic seismic section was generated by convolving the source wavelet with a randomly generated reflection coefficient section. Then, the noise extracted from the East sea SO data was added. Fig. 16 (a) shows the generated synthetic water column reflection section, and Fig. 16 (b) shows
the noise added section. We applied the trained D1 model and D2 model to attenuate the noise, and the results are in Fig. 16 (c) (D1 model) and (d) (D2 model). Most of the noise was successfully attenuated, but the noise was not perfectly removed in the D1 model result at a distance from 20 to 25 km and depth from 140 to 180 m. Fig. 17 shows the calculated data slope spectra. The data slope spectrum of the noise-added section follows a $k_x^2$ slope, which is the slope of the random noise. After the noise attenuation, the data slope spectrum of the D2 model result (red line) follows the data slope spectrum of the noise-
free section (greed line) almost identically. The data slope spectrum of the D1 model result (blue line) does not follow the noise slope, but the data slope spectrum is distorted compared to the noise-free data. The comparison of data slope spectra using synthetic data shows that the D2 model can recover the true data slope spectrum better than the D1 model.

## 4 Application to the East Sea SO data

The DnCNN models trained with training datasets 1 and 2 (the D1 and D2 models, respectively) were applied to the East Sea SO data. We applied the trained DnCNN models to the seismic sections from 0.03 to 0.28 s (approximately 22.5 to 210 m) where the reflections exist

  Fig. 18 shows the results of applying the DnCNN to line 1. Fig 18 (a) is the line 1 seismic section from 0 to 0.28 s before the noise attenuation. The seismic section shallower than 0.03 s is dominated by noise from direct waves, which is muted at the

data processing stage, and the section deeper than 0.28 s mainly contains random noise. Fig. 18 (b) and (c) are the denoised seismic section after applying the D1 and D2 model, respectively. In both results, most of the random noise was successfully removed, and the reflections became clearer. The strong random noise that occurred in the shallow part of the processed seismic sections was substantially attenuated, and the noise located between 150 and 200 km were also properly removed. Since noise was successfully attenuated, reflections that were difficult to distinguish due to a low S/N ratio were improved. In particular,

the weak signals between 0 and 50 km and between approximately 0.1 and 0.18 s became clearer after noise attenuation. Fig. 18 (d) and (e) are the estimated noise using the D1 and D2 model, respectively. As shown, both models successfully discriminated the noise component from the reflections; thus, the estimated noise sections are almost identical to the noise component of the processed seismic section. Although both models successfully attenuated the noise in the seismic section of line 1, there are several differences. Reflections are not observed from 150 to 200 km and at approximately 0.2 s in the line 1

seismic section. The result from the D1 model still contains noise in that part, while the result from the D2 model contains lower noise levels compared to that from the D1 model.

  Fig. 19 shows the results of applying the DnCNN to line 2. Fig .19 (a) shows the line 2 seismic section from 0 to 0.28 s before the noise attenuation. Fig. 19 (b) and (c) show the denoised seismic section after applying the D1 and D2 model, respectively. The seismic section of line 2 was contaminated by severe noise, but the D1 and D2 model properly removed the noise. In

particular, the strong random noise located between 0 to 50 km was removed; thus, it became possible to recognize the reflections that were illegible. In addition, the reflections with steep slopes between 240 and 260 km and between 0.12 and 0.2 s were obscured by severe noise, but the D1 and D2 models successfully attenuated the noise and clearly recovered the reflections. However, similar to the line 1 result, the D2 model attenuated the noise better than the D1 model in some parts of the section. From 20 to 50 km, noise can still be observed when the D1 model is applied, but most of the noise has been

sufficiently suppressed when the D2 model is applied.

  Despite the successful noise attenuation of the D1 and D2 models, we found some differences. We presume that these differences are caused by the characteristics of the SEZ data which are the ground truth used to train the D1 model. The SEZ data are field data and contain noise to a certain degree because it is almost impossible to perfectly remove the noise from the field data. In other words, the D1 model is likely to regard the noise in the seismic section with similar characteristics to those

contained in the ground truth as a signal rather than noise. On the other hand, the D2 model does not suffer from this kind of problem because its ground truth is noise-free synthetic data.

To validate the noise attenuation results, we also calculated and compared the data slope spectra by using the outcome of the D1 and D2 models. Before calculating the data slope spectrum, we scaled the seismic sections again by multiplying the signal by the square root of time at each time step (consequently multiplying the seismic signal by the time at each time step) for the spherical divergence correction. Then, we converted the seismic section from the time axis to the depth axis using a constant sound speed of 1500 m s$^{-1}$ and extracted the part from 150 to 175 km and at a depth from 75 to 150 m. Fig. 20 (a), (b) and (c) show the seismic sections extracted from the section before and after noise attenuation using models D1 and D2, respectively. The seismic section before noise attenuation was severely contaminated with random noise, but most of the noise was removed in the sections after noise attenuation. Fig. 20 (d) shows the calculated data slope spectra. From the KM07 model (Klymak and Moum, 2007), noise has a $k_x^2$ slope in the slope spectrum, and we plotted the $k_x^2$ slope with the green dashed line in Fig. 20 (d) for comparison. The data slope spectrum of the section before noise attenuation has a $k_x^2$ slope at wavenumbers above 0.002 cpm, which indicates that noise dominates these wavenumbers. Because of the severe noise, it is impossible to analyze the seismic data before noise attenuation. On the other hand, the data slope spectra after noise attenuation seem to contain internal waves subrange from 0.0015 to 0.006 cpm and turbulence subrange from 0.009 to 0.015 cpm that approximately follow the $k_x^{-1/2}$ (yellow dashed line) and $k_x^{1/3}$ (purple dashed line) slopes (Klymak and Moum, 2007), respectively. This result indicates that noise was properly attenuated and the seismic data could be analyzed, even though noise with a slope of $k_x^2$ still occurred at wavenumbers above 0.02 cpm. There is a shift in the data slope spectrum after noise attenuation at wavenumbers smaller than 0.001 cpm. This shift is also observed in the synthetic data slope spectrum experiments. In Fig. 17, there is a difference between the spectrum of the noise-added section and that of the noise-attenuated sections at wavenumbers smaller than 0.001 cpm. However, the difference is also observed between the spectrum of the noise-free section and that of the noise-added section. Therefore, this shift seems to be caused by the characteristic of the noise extracted from the East Sea SO data.

From the noise attenuation results obtained by applying the trained models to the East Sea sparker SO data, we showed that the DnCNN architecture used in this study can successfully suppress random noise. The comparison of the D1 and D2 model results showed that the training data generated using noise-free synthetic data are more suitable for random noise attenuation of sparker SO data than those generated using field data with a relatively high S/N ratio.

**5 Summary**

Random noise is one of the major obstacles in analyzing SO data. Conventionally, the noise in SO data has been attenuated through simple data processing methods because most of the SO data are obtained with air guns, which generate data with a high S/N ratio. However, the simple noise attenuation method is not sufficient for data with a low S/N ratio, such as sparker SO data. Despite the low S/N problem, the sparker source has advantage of generating a higher-frequency band signal than an air gun source and can provide information with finer vertical resolution. Therefore, we applied machine learning to attenuate the random noise in East Sea sparker SO data, which contains significant random noise. The DnCNN architecture was used to construct a neural network, and training data were generated by combining the ground truth and noise extracted from the target seismic data at random amplitude ratios. Two different training datasets were generated, and they used either field or synthetic data as the ground truth. The trained DnCNN models were applied to the test datasets that were generated with the same procedure of generating the training datasets. The test results were verified based on the PSNR, SSIM, trace extraction and normalized RMS error. The data slope spectrum test using synthetic seismic section was also performed. The test results revealed that both trained DnCNN models were able to successfully attenuate random noise and the training data generated using noise-free synthetic data showed better results than the training data generated using high-S/N ratio field data. We applied the trained DnCNN models to the East Sea sparker SO data, which is the target of this study, and the models successfully attenuated random noise. The comparison of the denoised seismic sections after applying the two different trained models also showed that the training dataset generated from the noise-free synthetic data was more suitable for sparker SO data noise attenuation than that generated from the high-S/N ratio field data.

Although random noise is almost completely attenuated in the seismic section, the proposed method still needs several improvements. First, the calculated data slope spectrum indicates that a noise with a slope of $k_x^2$ is not removed completely at wavenumbers above 0.02 cpm. Therefore, future studies should include a detailed analysis of the slope spectra of the SO data and establish an improved noise attenuation algorithm suitable for higher wavenumbers. Moreover, the data were collected and processed using 2D seismic exploration technology, which cannot efficiently deal with out-of-plane contamination. We expect that 3D seismic exploration can improve the resolution of SO data.

The network architecture used in this study is straightforward and efficient. In addition, the proposed method of generating the training dataset is very simple and easy because it only requires synthetic data, which are readily generated, and noise data, which can be extracted from the target seismic data. Moreover, only approximately one hour is required to train the DnCNN model with a single GPU. Therefore, the noise attenuation method suggested in this study can be widely and easily applied for noise attenuation of the various kinds of SO data.

**Data availability**

The code, synthetic training data samples, field noise data are available at https://www.doi.org/10.5281/zenodo.4020335. Marmousi 2 model is available at https://wiki.seg.org/wiki/AGL_Elastic_Marmousi, Sigsbee2A model is available at http://www.delphi.tudelft.nl/SMAART/,
and 1994 BP statics benchmark model is available at https://wiki.seg.org/wiki/1994_BP_statics_benchmark_model. The East Sea sparker field seismic data can be made available upon request to authors.

**Author contribution**

Hyunggu Jun and Hyeong-Tae Jou constructed the machine learning program and performed experiments. Chung-Ho Kim, Sang Hoon Lee and Han-Joon Kim acquired seismic data and performed data processing.

**Competing interests**

The authors declare that they have no conflict of interest.

**Acknowledgements.**

This research is supported by the Korea Institute of Ocean Science and Technology (Grant numbers PE99841 and PE99851) and the Korea Institute of Marine Science and Technology promotion (Grant number 20160247).

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

Figures

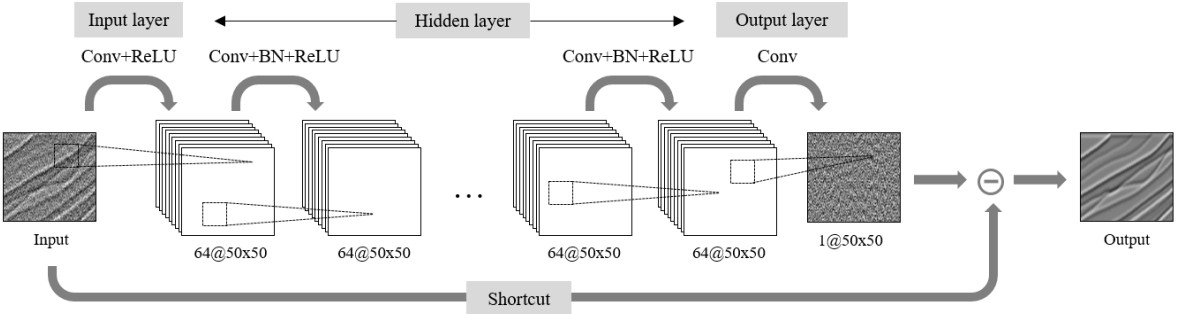

Figure 1: DnCNN architecture. 64@50x50 indicates 64 feature maps with 50x50 size, Conv is two dimensional 3x3 convolution kernel, BN is batch normalization, and ReLU is rectified linear unit activation function.

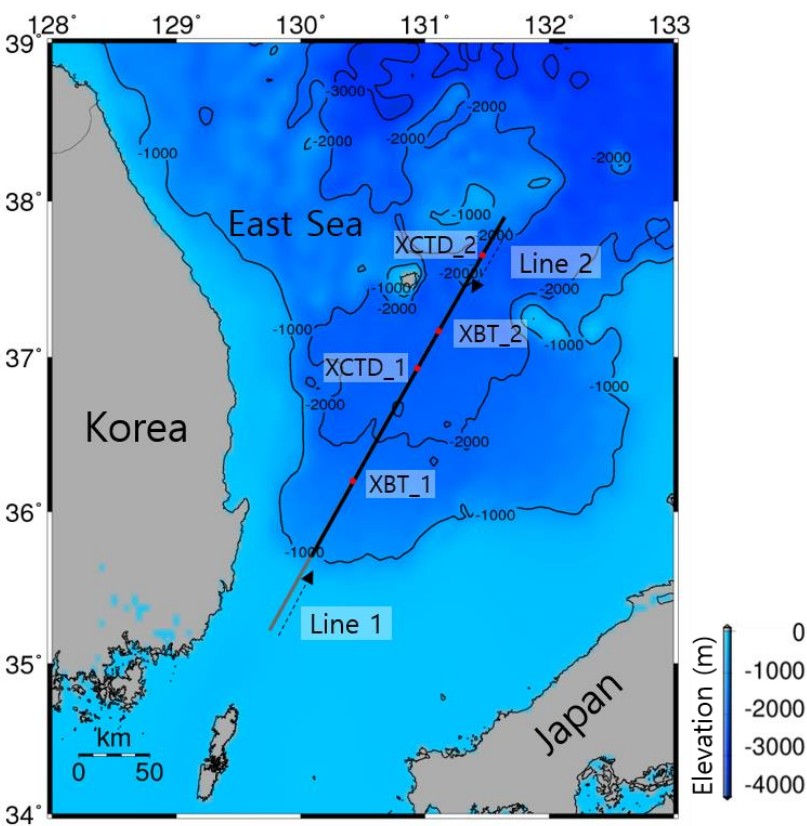

Figure 2: Location of seismic exploration. The gray and black solid line is the survey line. The gray line indicates the shelf and slope parts, which were removed from the seismic section during data processing, and black line indicates the target area of this study. The black dashed lines with arrows indicate the exploration directions of lines 1 and 2, and red dots are the locations of XBTs and XCTDs.


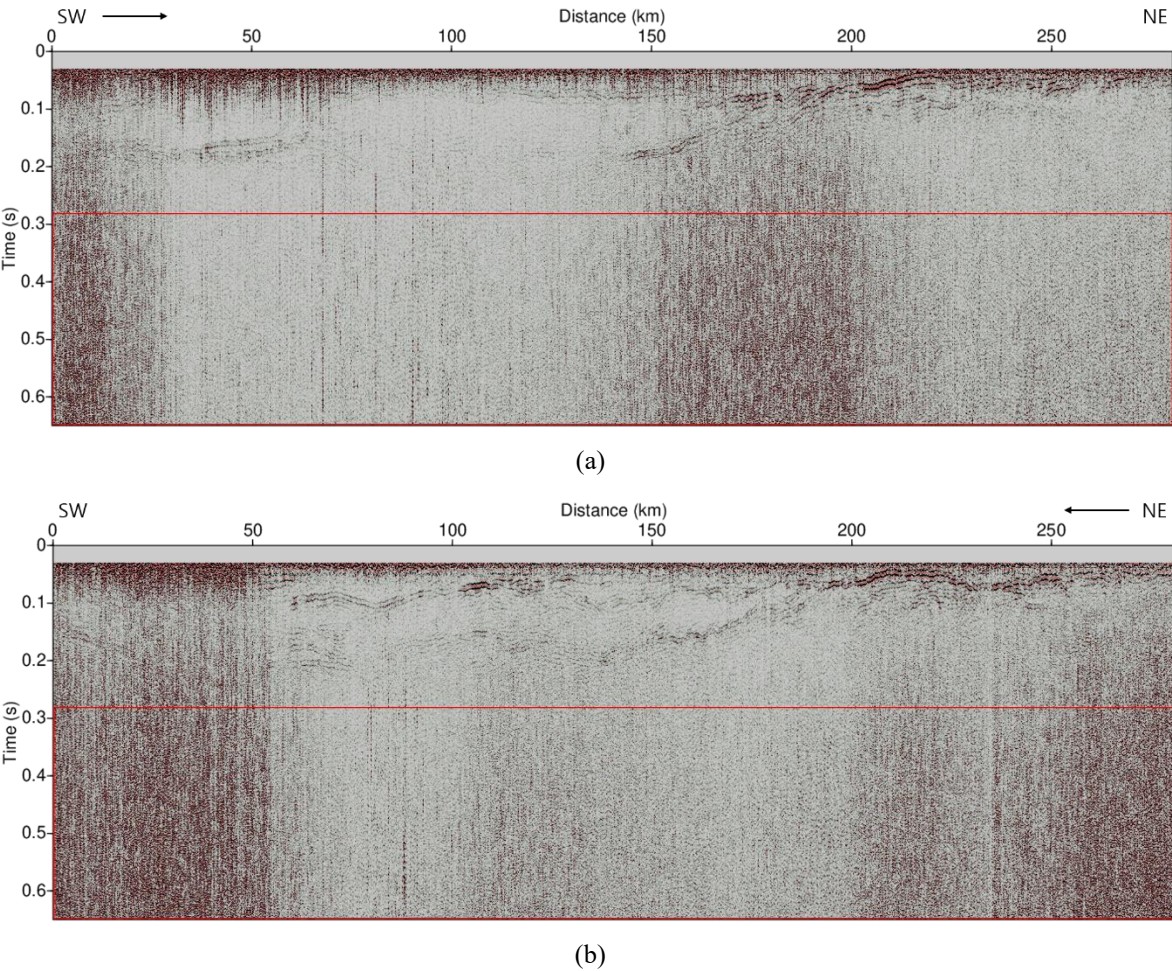

Figure 3: Processed seismic section of the East Sea: (a) line 1 and (b) line 2. The seismic section in the red rectangle is the noise part used to generate the training data. SW indicates south west, NE indicates north east, and the black arrow indicates the data acquisition direction.

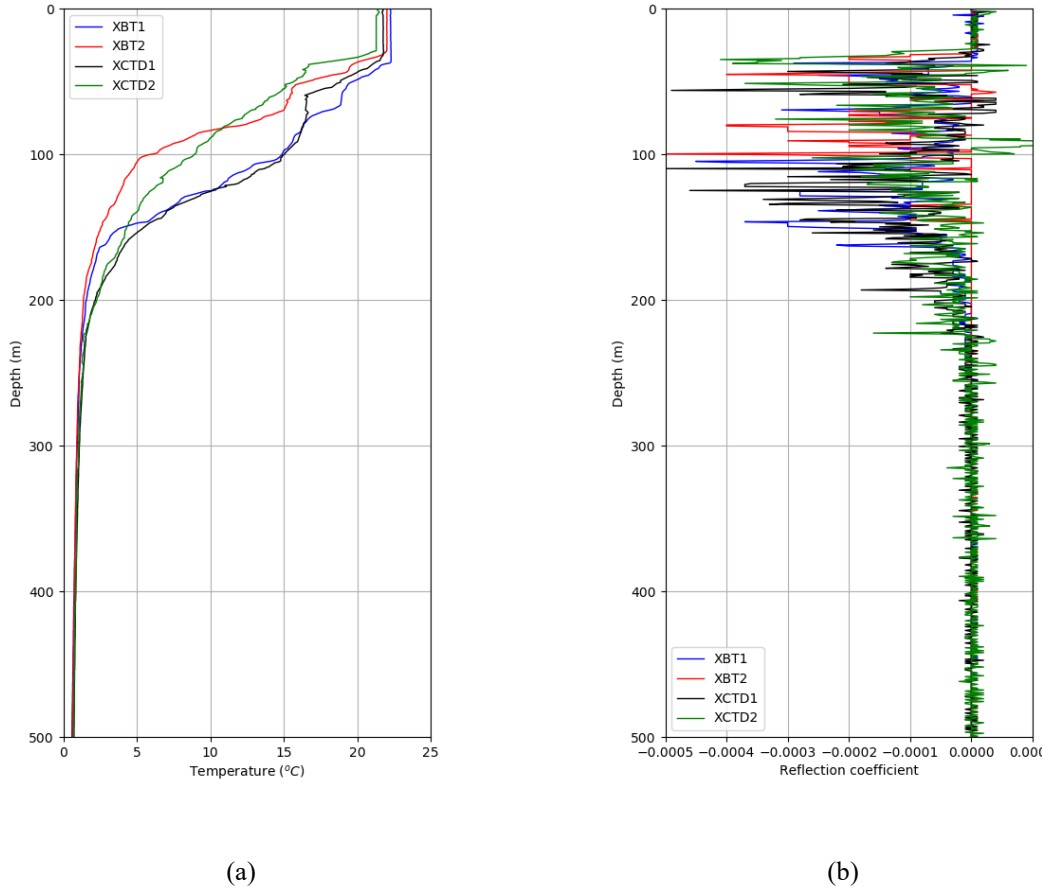

<table>
<tr><td>(a)</td><td>(b)</td></tr>
</table>

Figure 4: (a) Temperature and (b) reflection coefficient profiles obtained using 2 XBTs and 2 XCTDs.


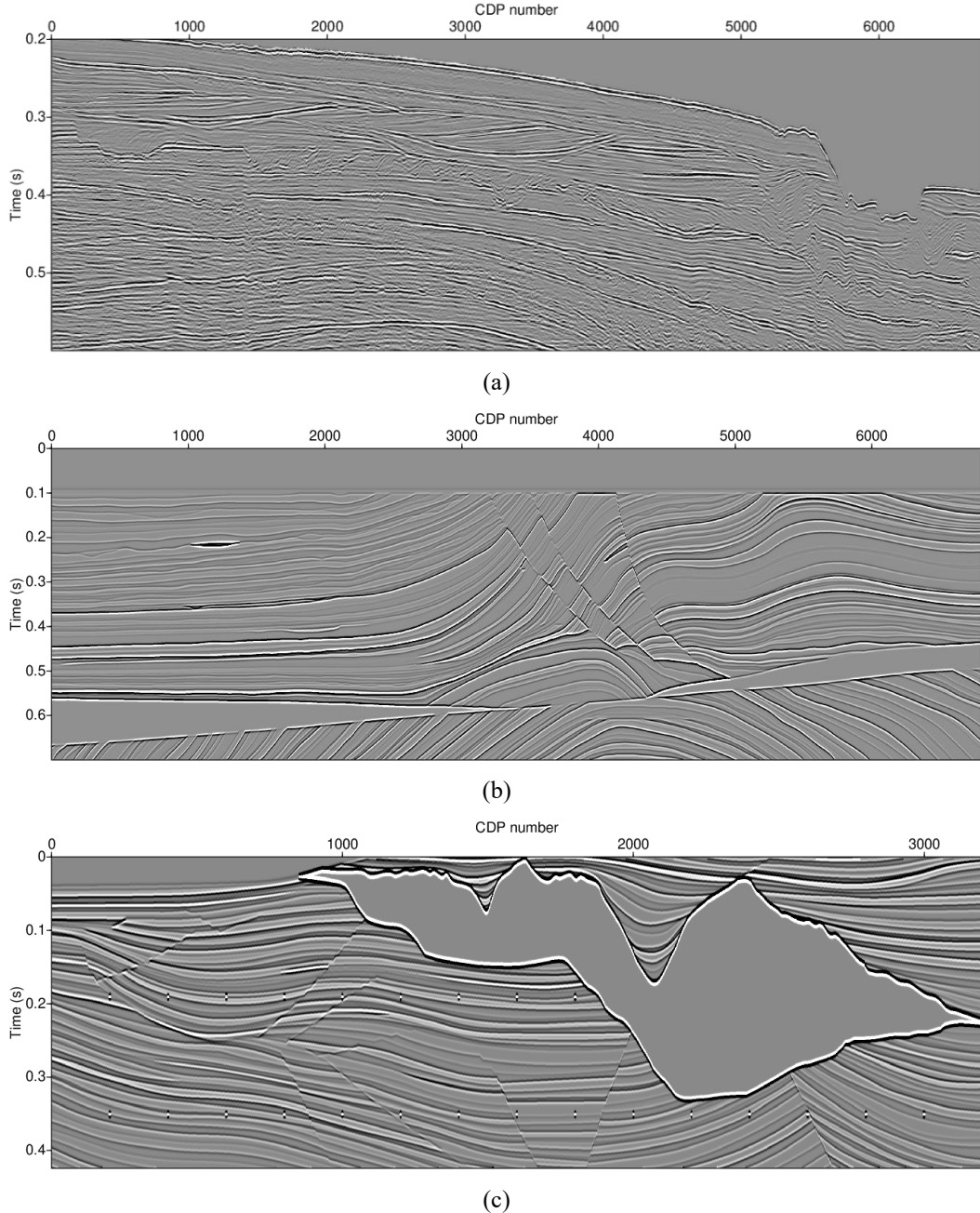

Figure 5: (a) Processed SEZ field seismic section, (b) Marmousi-2 synthetic seismic section and (c) Sigsbee 2A seismic section used to generate the training data.

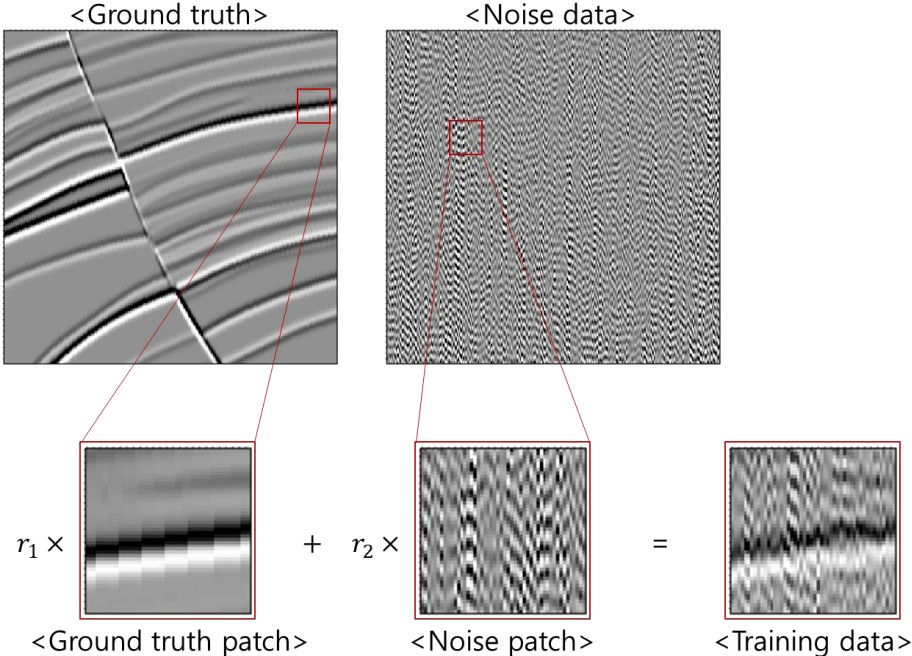

Figure 6: Example of constructing the training data.

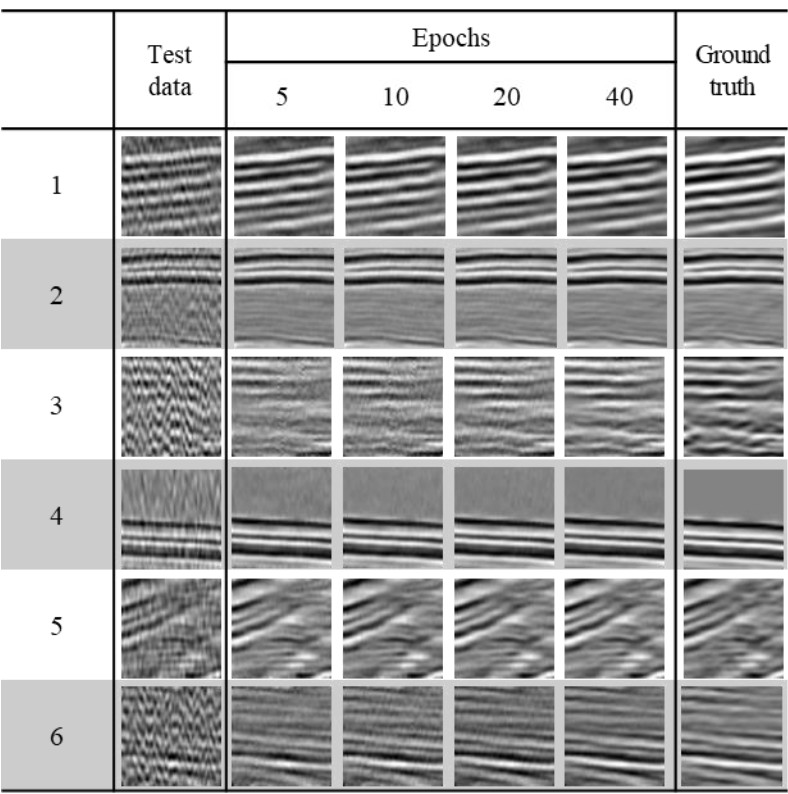

Figure 7: Test data, ground truth, and denoised results after applying the DnCNN models trained using training dataset 1.


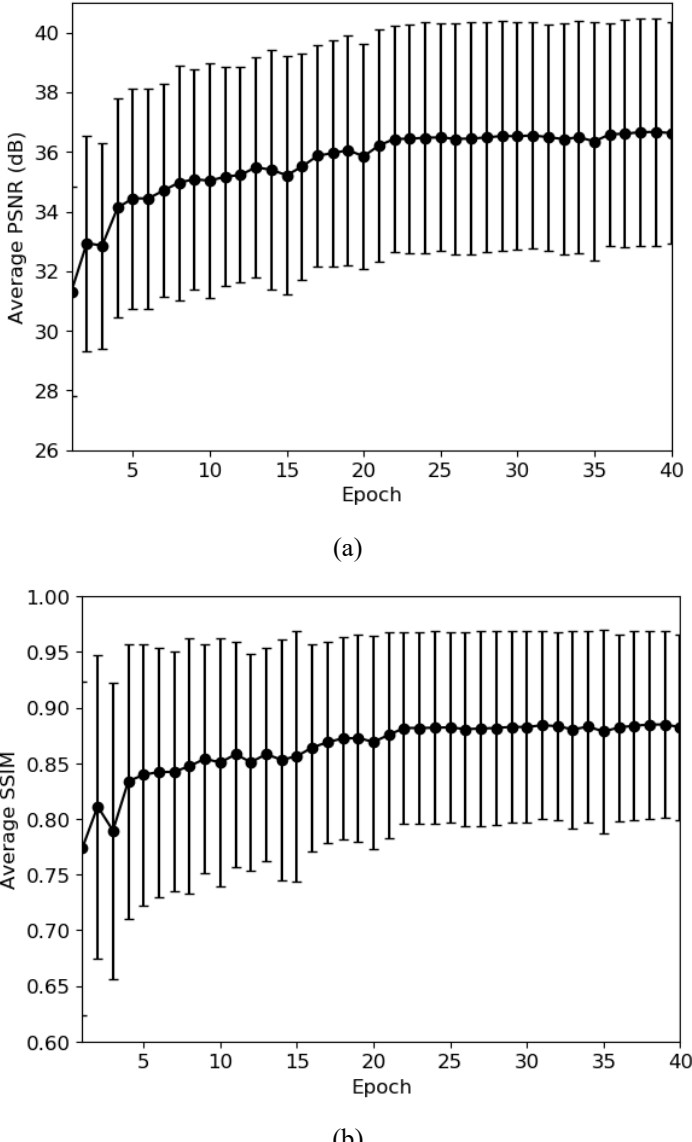

(a)

(b)

Figure 8: Average (a) PSNR and (b) SSIM with standard deviation of the test result of the first experiment.

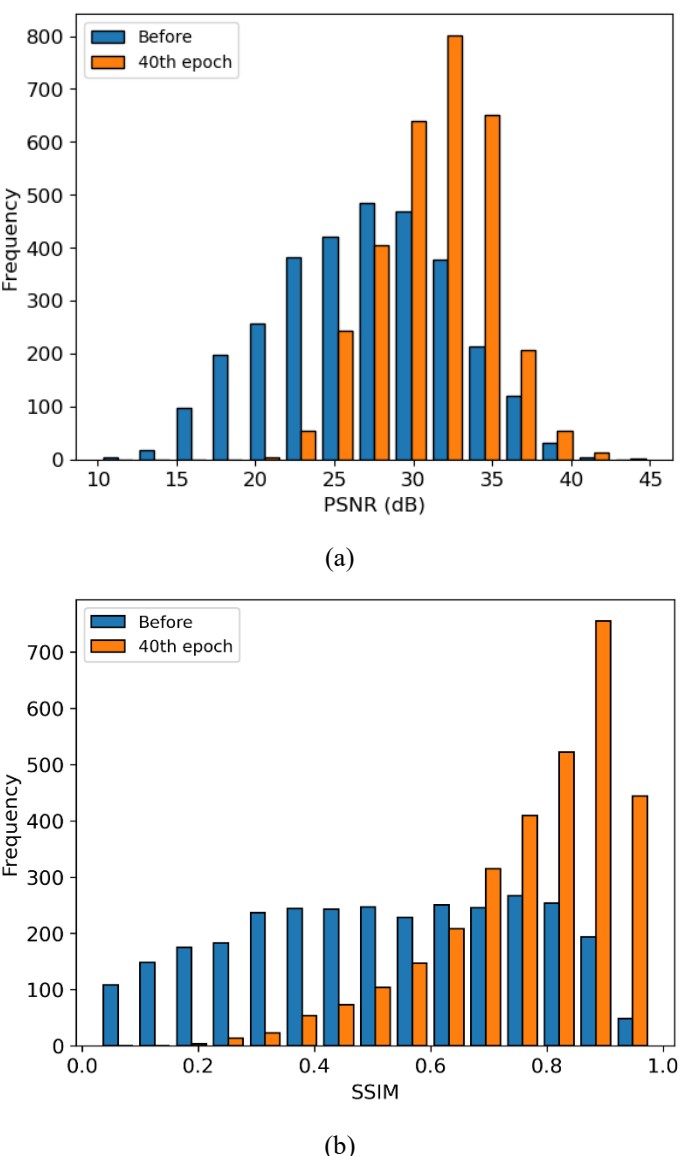

(a)

(b)

Figure 9: (a) PSNR and (b) SSIM histogram of the test data before applying the D1 model and after applying the 40th epoch of the D1 model.


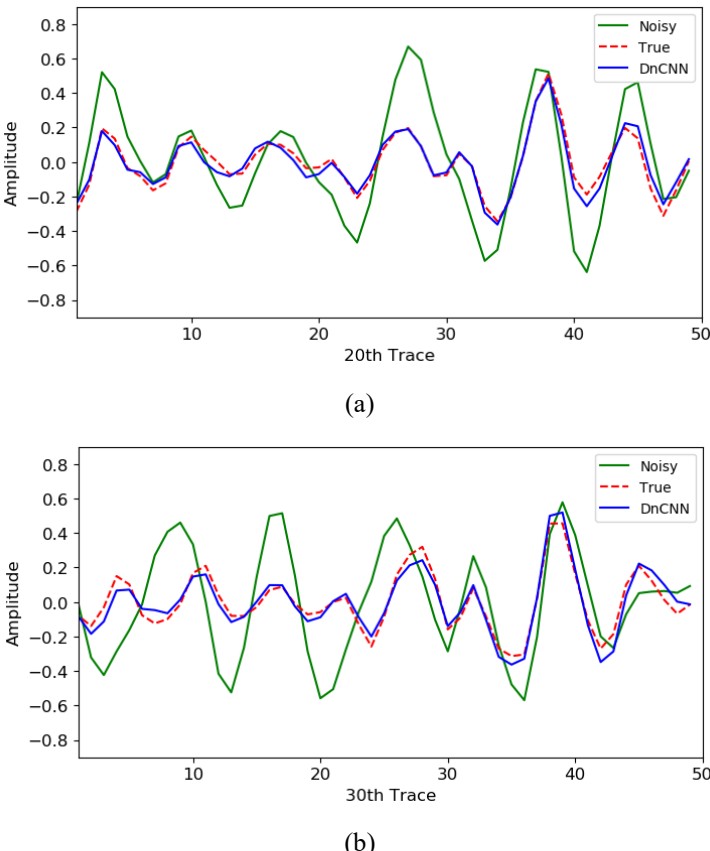

(a)

(b)

Figure 10: Comparison of the extracted traces before applying the D1 model and after applying the 40th epoch of the D1 model. The green solid line is the trace from the noisy data, the red dashed line is the trace from the ground truth, and the blue solid line is the trace from the denoised data after applying the D1 model. (a) is the 20th and (b) is the 30th vertical trace of the last test patch in Figure 7.

Figure 11: Test data, ground truth, and denoised results after applying the DnCNN models trained using training dataset 2.

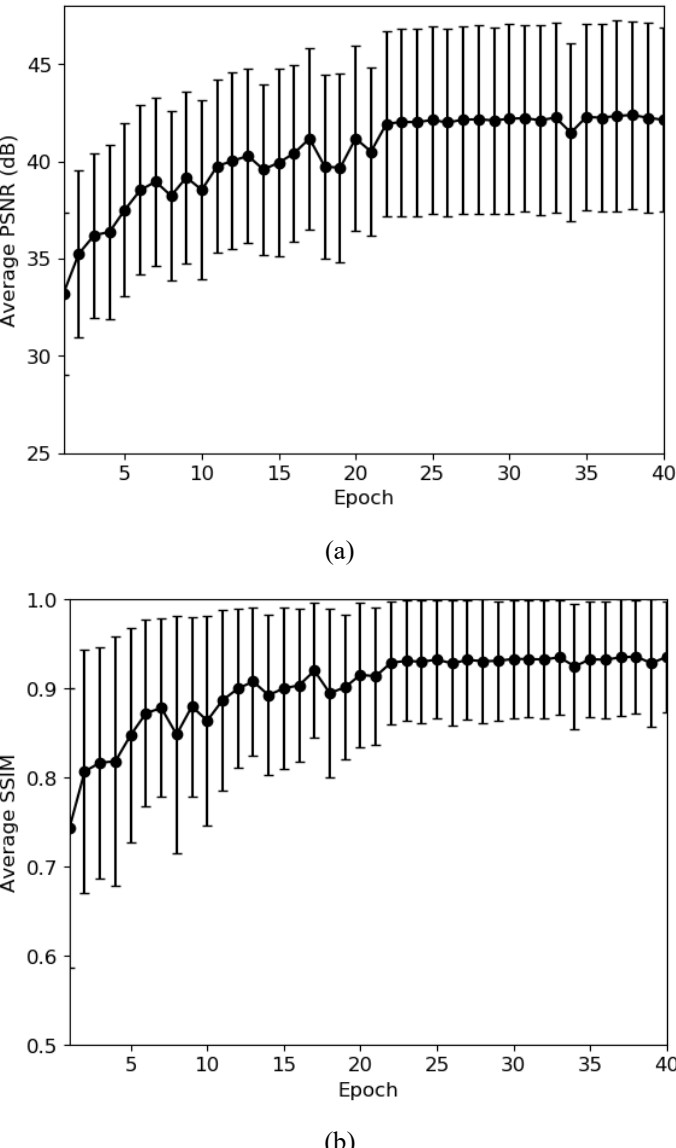

(a)

(b)

Figure 12: Average (a) PSNR and (b) SSIM with standard deviation of the test result of the second experiment.


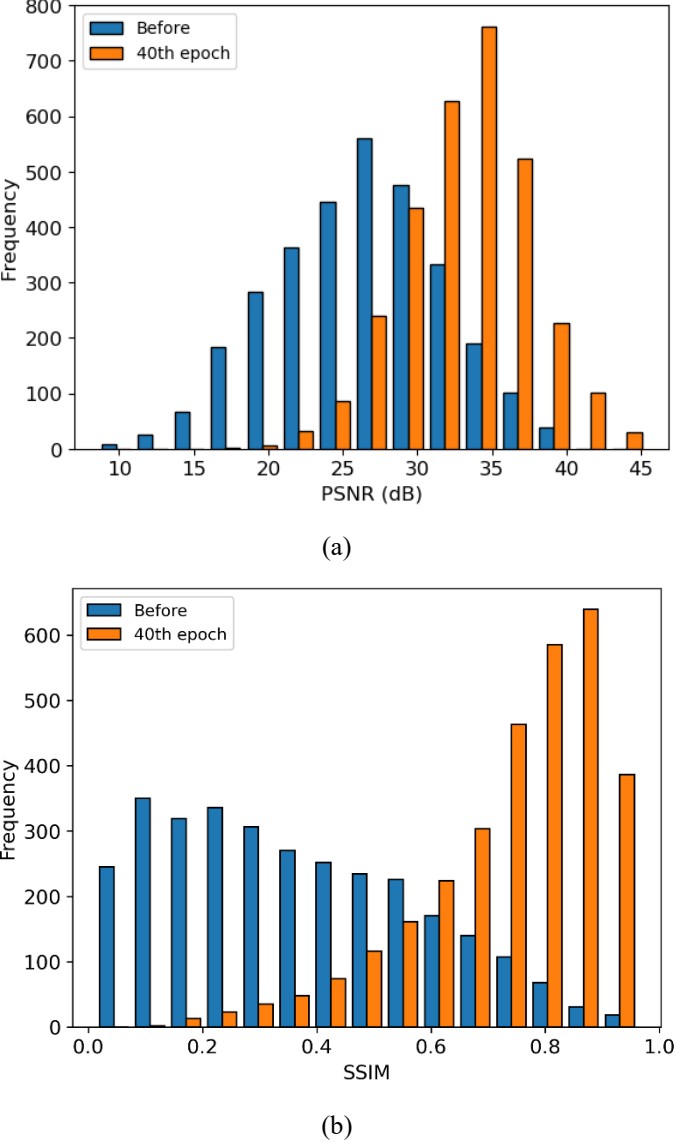

(a)

(b)

Figure 13: (a) PSNR and (b) SSIM histogram of the test data before applying the D2 model and after applying the 40th epoch of the D2 model.

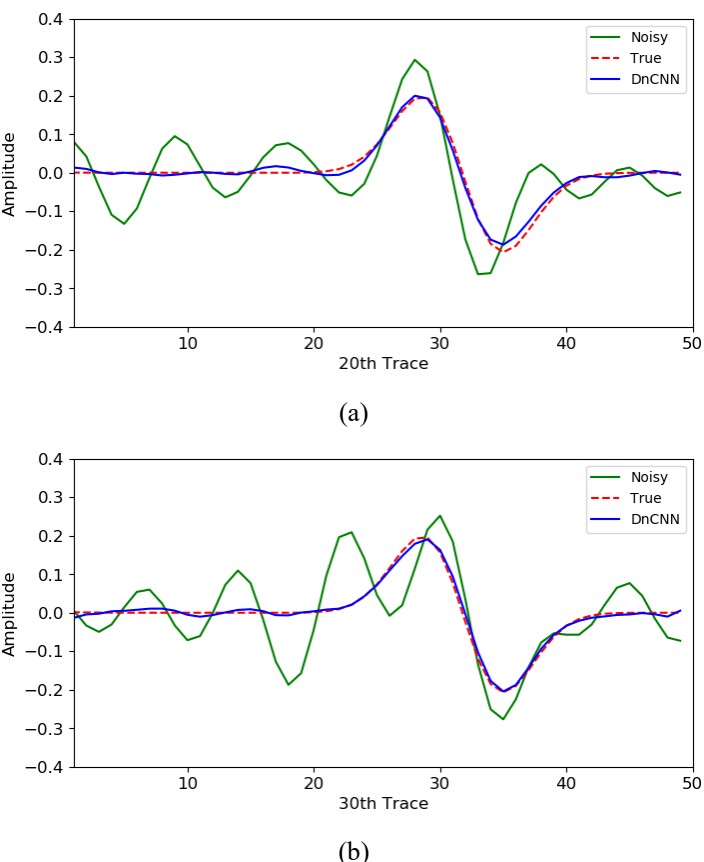

(a)

(b)

Figure 14: Comparison of the extracted traces before applying the D2 model and after applying the 40[th] epochof the D2 model. The green solid line is the trace from the noisy data, the red dashed line is the trace from the ground truth and the blue solid line is the trace from the denoised data after applying the D2 model. (a) is the 20[th] and (b) is the 30[th] vertical trace of the first test patch in Figure 11.


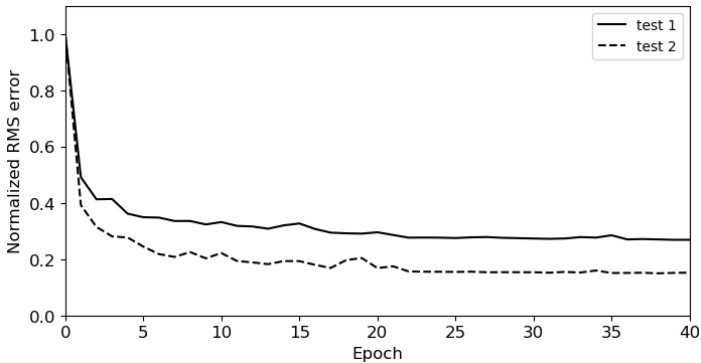

Figure 15: Normalized RMS error between the ground truth and denoised result of the first (solid) and second (dashed) experiments.

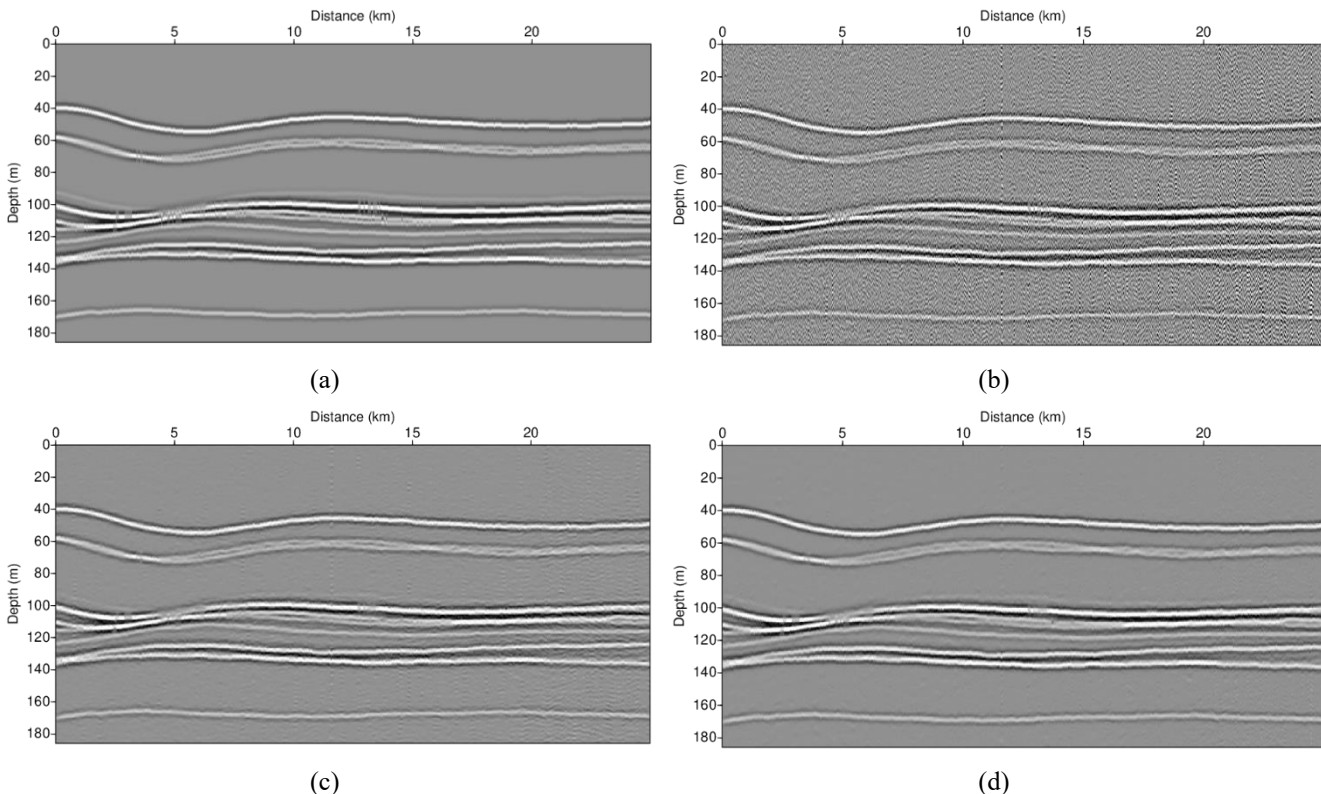

Figure 16: (a) Noise-free and (b) noise-added synthetic water column reflection section and noise-attenuated results using (c) the D1 model and (d) the D2 model.

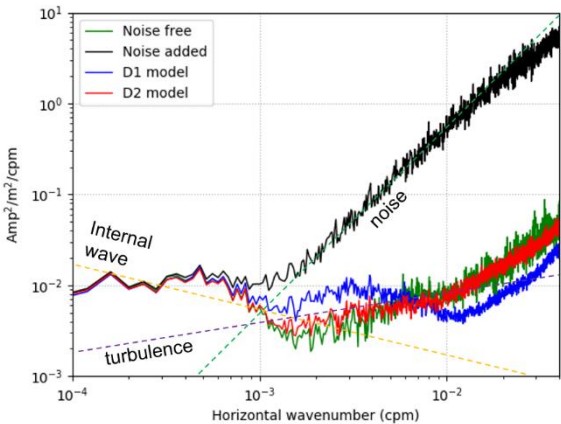

Figure 17: Data slope spectra of noise-free (green) and noise-added (black) synthetic seismic sections and noise-attenuated synthetic seismic section using the D1 model (blue) and D2 model (red).

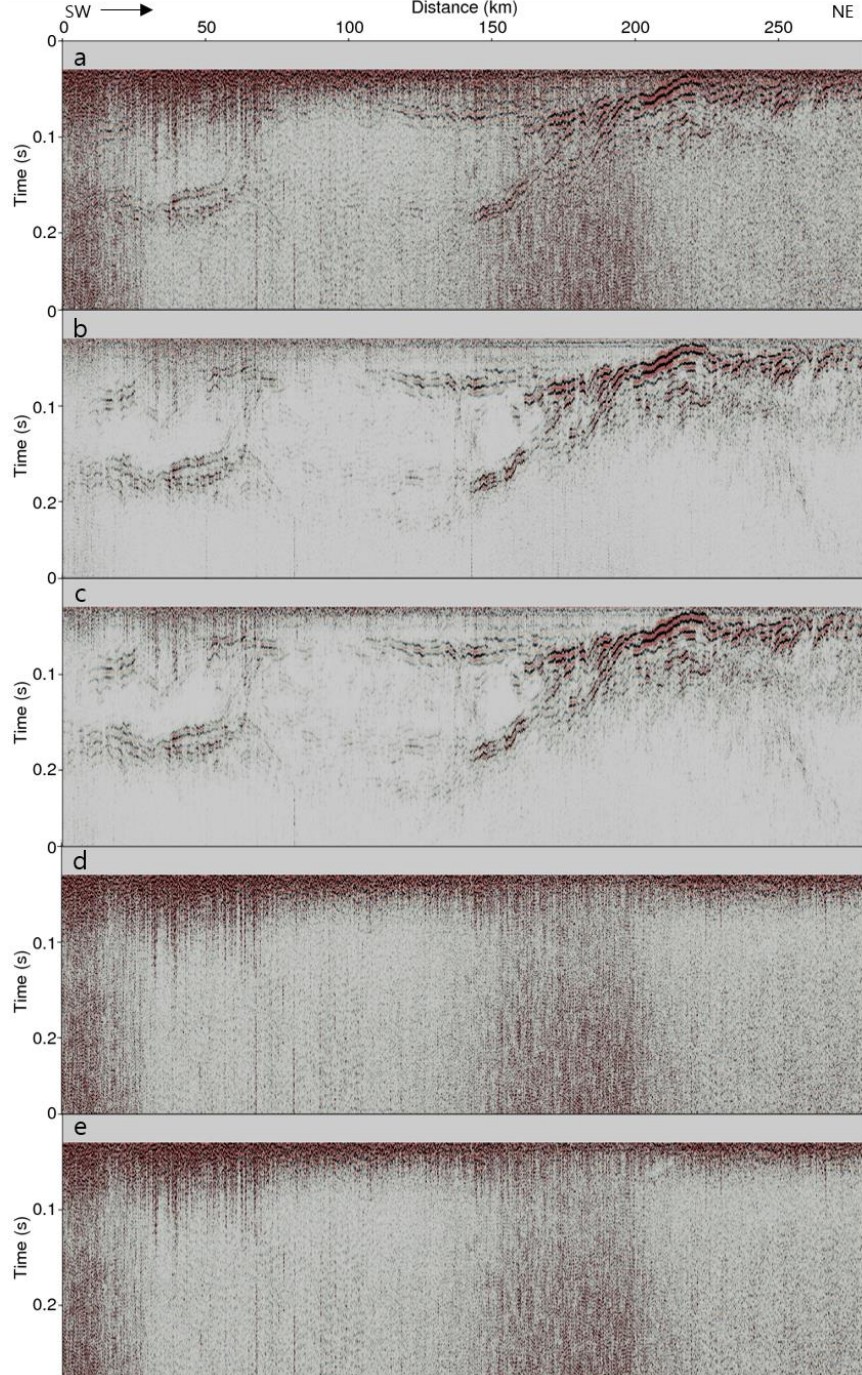

Figure 18: (a) Line 1 seismic section before applying DnCNN, noise-attenuated result using (b) the D1 model and (c) the D2 model, and estimated noise using (d) the D1 model and (e) the D2 model.


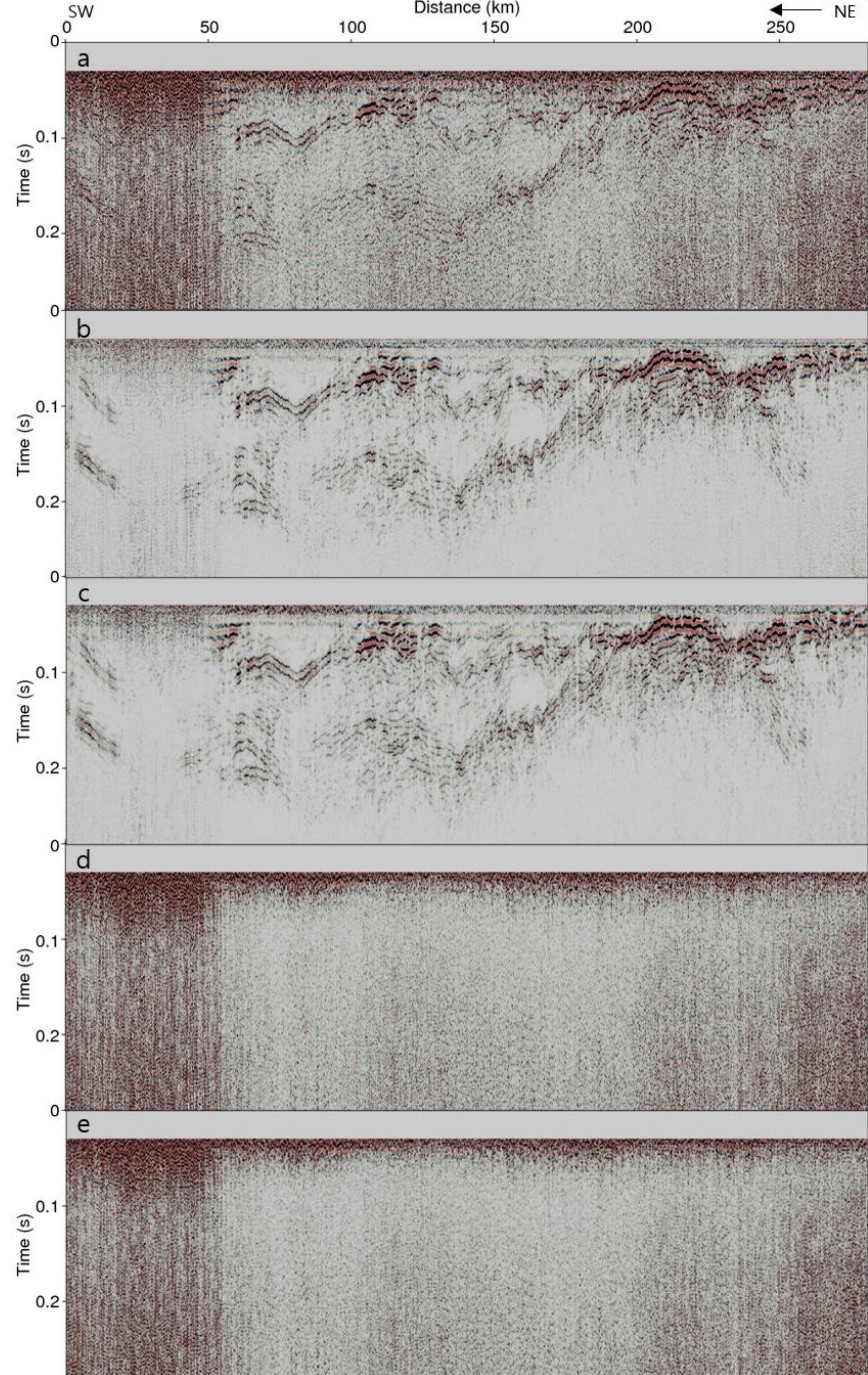

Figure 19: (a) Line 2 seismic section before applying DnCNN, noise-attenuated result using (b) the D1 model and (c) the D2 model, and estimated noise using (d) the D1 model and (e) the D2 model.

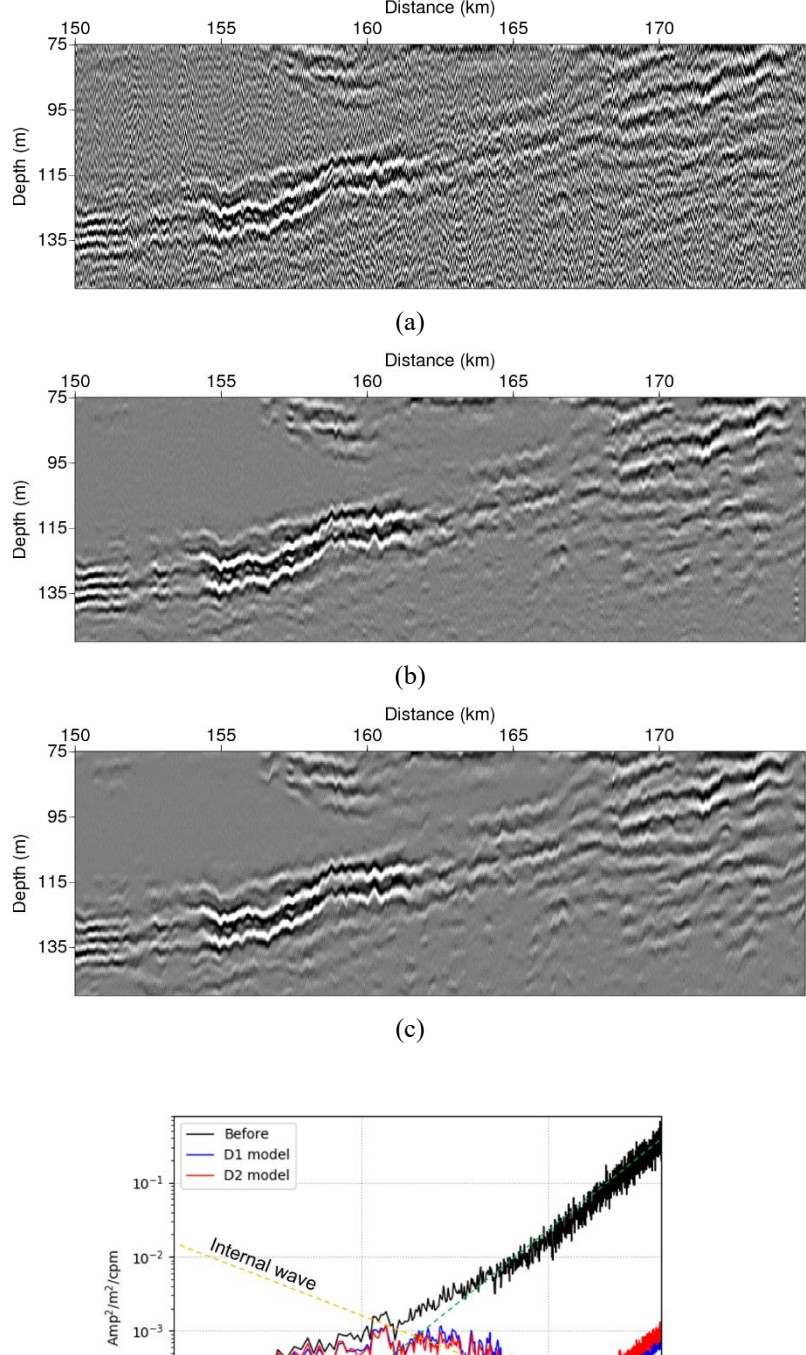

(a)

(b)

(c)

Figure 20: Extracted seismic sections ((a) is the section before noise attenuation, (b) is the section after applying the D1 model and (c) is the section after applying the D2 model). (d) shows the calculated data slope spectra of (a), (b) and (c).