# Peer review of "Random Noise Attenuation of Sparker Seismic Oceanography Data with Machine Learning"

_Ocean Science, 2020_

## Referee Comment (RC1) · Anonymous Referee #1 · 22 Apr 2020

This study applies machine learning method to extract the signal from noisy seismic signal from the water column collected using sparker source. This is an alternative means of collecting data (to air guns) offering lower costs despite larger noise in the signal. Authors offer a method that can successfully clean the data and help gain insight on physical oceanography. For the method, the training data are important, and the authors present cases using synthetic and field data.

Overall the study is technical, but can be of interest to the Ocean Science readers. Authors demonstrate convincingly that the random noise attenuation with machine learning satisfactorily cleans the data and reveals the oceanographic structure in the seismic

**signal.**

I have a list of detailed comments below. These are mainly clarifications and suggestion for better presentation. The presentation of the material in figures is poor and needs improvement. The analysis is solid and does not need any revisions in my opinion (non-expert on this technique). However, I am surprised the authors do not give a more thorough analysis of the error and skill (e.g., by histograms using all 3072 data patches instead of only 6).

The context in an oceanographic application comes finally in the last figure and the last page of the text (line 364 and on). I would strongly encourage the authors to improve the "Ocean Science" part by adding more insight using the clean seismic sections. Can you make some oceanographic inferences, interpretations (or better quantification) using the cleaned Lines 1 and 2?

Finally, it would serve the community much better if the authors made available some code for noise attenuation using machine learning. They offer the code through communication with authors, but the impact would be far larger if they make the code available as a supplement.

**Detailed comments:**

Li 6-7. The opening sentence of the abstract is a bit confusing. SO exploits water column reflections to interpret the oceanic features (fronts, eddies, water mass bound-aries) as well as ocean fine structure (internal waves etc.). Futhermore, "compensating for the drawbacks of conventional PO equipment" is a very strong (and erroneous) statement. Perhaps "supplements the conventional PO observations". We should also be very careful when interpreting the seismic images to describe PO quantitatively.

Li 8. The low / high frequency band introduction is not very helpful unless you relate it to spatial resolution.

Li 10. Reword "To solve the problem"? For example, "To extract reliable signal from the
low S/N ...."

Li 23: "measurements [from cruises] are performed..... observation [stations]."

Li 27: mention how the sea water characteristics can be estimated (through the acoustic impedance contrasts and expand a bit more to inform the reader)

Li 29-32: Fine, but please do not oversell. Perhaps mention "qualitative images" and then move to "quantitative information after careful analysis where temperature/salinity contrasts produce well-defined horizons of seismic reflections" or similar. Also SO is not "widely used".

Li 32: reword "determine the behavior of turbulence and internal waves" to, for example, "quantify the internal wave spectral distribution and infer turbulence"

Li 33: clarify what central frequency is (since the source covers a range of frequencies)

Li 43: vertical resolution of 1.5 m is not much superior to the vertical resolution of "several meters" stated in line 34. Perhaps specify the latter as 5-10 m?

Li 67-68: If not using MLP and AE (and any other acronym), no need to introduce them. It is difficult to read the text.

Li 64-75: If there's a possibility to thin out various methods introduced (and refer to a few key references and citations therein), it can be easier for the reader to follow.

Li 77: East Sea appears very abruptly here, out of context.

Li 135: delete "On the other hand,"?

Sec 2.1 and 2.2: can any of these descriptions refer to Fig 1? (I only see a reference in the end, at li 141, and it is not very instructive.)

Li 147: This is actually one line, but two repeats (in different travelling directions). Please mention the date of data collection, vessel speed during data collection. Transect duration etc.

OSD
Li 163-164: there're CTD /XCTD profiles, but the authors shown only 2xtemperature profiles from XBTs. It would be nice to increase the oceanographic context in the paper.

Li 167: please describe what a reflection coefficient is.

Li 184-185: what do you mean by "thus, the subsurface seismic data have a better S/N ratio than the SO data."? Is subsurface seismic data not SO data? I suspect you mean beneath seabed by subsurface. Please clarify.

Li 187: It is confusing: "We used the interval from 0.2 to 0.6 s of the original data where the noise level is relatively low". Earlier you mentioned that part was just noise!

Li 190: Reword "the data are field data recorded with the same equipment." as "the data are collected by the same equipment"

Li 204: what is g/cc? Please use SI units.

Li 249: bottom right (instead of right bottom)

Li 249-250: The sentence is confusing: "...using training dataset 1 has one problem. The ground truth of test data 5 contains noise in the right bottom part, and training dataset 1 also contains noise in some parts of the ground truth". Dataset 1 has 6 test data. With the last reference to dataset 1 do you mean test data 1 or the entire dataset 1? Perhaps cut out the entire last part after the comma. Overall, I would appreciate a more distinct wording for test data. For example subset 1 to 6, or patch (you use it in line 280)?

Li 279-280: 20th and 30th traces from the last patch: which epoch is this? Are the traces from the 50x50 patch? Can you please mention for the reader: "...traces out of the 50x50 size patch 6 of the test data".

Li 310-311: can be cut out; simply cross reference Fig 13 after 25 epochs. Overall there are repetitions throughout the authors could try to simplify.

Li 325: perhaps specify, "is the number of test data patch (3072)"
Li 327-328: too many significant digits at RMS errors? (perhaps enough with 6.37 and 6.34). For which epoch are these values? (Also the normalized values in line 331 could be 0.27 and 0.15)

Li 332: delete "than that of the D1 model"

Eq 1, is a division by nmode missing?

Li 364: "The data slope spectrum is the slope spectrum..." this is all very confusing. The data slope spectrum is first referred to in line 276-277 (again without explanation). Please introduce what the data slope is. For example, "the slope spectrum is the horizontal wavenumber,  $k_x$ , spectrum of the horizontal gradient of the vertical displacement of a digitized horizon. The data slope spectrum is ...?" (or a similar explanation. Note my interpretation of the slope spectrum can be in error.)

Li 367: replace "we calculated the data slope spectrum .... and compared the data slope spectra" with "we calculated and compared the data slope spectra using the outcome of the D1 and D2 models...."

Li 376: "slope" is missing before "at wavenumbers"

Li 377-378: I cannot quite follow the subranges and the mentioned slopes in this panel. Perhaps mark on the figure?

Li 390: Here again mention why sparker SO data may be desirable

Fig 1. Please offer some more explanation in the caption. If not possible, defer reader to the main text.

Fig 2. Elevation is grayed out for >0m, so the colorbar can stop at 0. It would be useful to add a few isobaths. I would call Line 1 and Line 2, Repeat 1 and Repeat 2.

Figs 4 and 5 can be combined into 1 figure. I suggest two panels, T profiles in one panel with different color. Reflection coeff in the second panel with different colors and one offset by 0.0001 unit. Does the coefficient have a unit?
Interactive

comment

Fig 6 can be removed. It is already shown in Fig 3 and with the statement time > 0.28 s. You can mark the region by a rectangle in Fig 3.

Fig 10 (and Fig 13)- this is the average PSNR and SSIM for the 6 subsets of dataset 1? Would it not be better to show all 6 lines, or the average with one standard deviation? Actually the number of test data is 3072 (6 is an arbitrary pick), why not show the mean and std over all 3072? And also show a histogram?

Fig 11 caption: in the end the cross reference should be to Fig 9.

Fig 14 caption: in the end the cross reference should be to Fig 12. In the text the model is referred to as D2 (but here D1).

Fig 16. This figure is not needed either. It is simply the upper 0.28s of Fig 3. However, I appreciate that it is zoomed in and compared to the cleaned sections. See suggestion below Fig 17 comment.

Fig 17. Please consider removing xaxis labels from panels c to d, and placing panel labels in the upper left corner of panels (it's grayed out anyway), so that you can have a more condensed 4-panel, 1 page figure with minimum white space vertically between panels.

I think a reorganized version of Figs 16- 18 will be much better for the reader. I suggest 2 figures, each with 5 panels (with identical x-axis limits and width, and minimum white space between them). New Fig 16: Line 1 results. New Fig 17 Line 2 results with corresponding 5 panels for each figure:

a) data (as in Fig 16a)

b) clean after D1

c) clean after D2

d) noise after D1

OSD
e) noise after D2.

It is most impactful, if you can fit all 5panels in one page please.

---

## Referee Comment (RC2) · Richard Hobbs (Referee) · 27 Apr 2020

The paper describes a signal processing algorithm that can be used to suppress noise in images and how it can be applied to seismic oceanography data, in particular sparker data which inherently has a higher frequency content than air-gun data but has a lower signal-to-noise ratio. The algorithm uses a Neural Network to identify and suppress the noise. The authors present a fair and balanced assessment of the method, highlighting the issues with training the Neural Network to discriminate what is wanted from what is not wanted. In this case the training is to extract the noise which is then subtracted from the original data to reveal the underlying reflectivity.

[Figure]

As it is written I think the detailed description of the methodology will be of low interest to the majority of readers of this Journal. However, the paper does show how SO may be used to successfully investigate the turbulent sub-range and fine-scale mixing. So I am recommending the authors revise the paper: the detailed methodology and training should be moved to supplementary material; then the paper should focus and expand on the East Sea data and its analysis of the implications for understanding ocean processes.

Richard Hobbs

The English is generally poor as is sentence structure, however despite this the meaning is generally unambiguous. Recommend careful editing to improve readability.

For example I suggest the authors consider the revised Abstract below:

Abstract. Seismic oceanography (SO) acquires water column reflections using controlled source seismology and provides high lateral resolution that enables the tracking of the thermohaline structure of the oceans. Most SO studies obtain data using air guns, which can produce acoustic energy 0ver the 5-150 Hz bandwidth. For higher-frequencies other seismic systems may be used, such as a sparker source with central frequencies of 250 Hz. However, the sparker source has relatively lower energy compared to air-guns and consequently produces data with a lower signal-to-noise (S/N) ratio. To address this problem we apply machine learning to attenuate the random noise without distorting the true shape and amplitude of water column reflections. Specifically we use a denoising convolutional neural network (DnCNN) that successfully suppresses random noise in a natural image. One of the most important factors of machine learning is the generation of an appropriate training dataset. We generate two different training datasets using synthetic and field data then the trained filters are applied to test data, and the denoised results are quantitatively compared. To demonstrate the technique, the trained filters are applied to an SO sparker seismic dataset acquired in the East Sea and the denoised seismic sections are evaluated and show

..... The results demonstrate that machine learning can successfully attenuate the random noise in sparker water column seismic reflection data.

Please use the above as an example of how to clarify your English but also how better to engage with your intended audience who are oceanographers. I do not intend to rewrite the rest of the paper for you! Further, this paper requires a significant revision so detailed correction at this stage serves no purpose.

line 34 - delete "relatively low" as you do not state relative to what. Please edit paper to remove, as much as is possible, unqualified comparative statements.

line 46 - This problem is more accentuated in SO because the impedance contrasts between the layers are small.

line 65 - this reference list ignores the long history of the use of Neural Networks see McCormack's paper in Leading Edge 1991 which shows an early attempt to use these NN to identify noisy traces in seismic data, since then NN have been evaluated for many tasks in the processing of seismic reflection data. Suggest authors change sense to recognise the history but equally highlight the recent advances in AI. I now note that this history is partly addressed in the following paragraph.

line 153 - scaling by the sq-rt of time is not "spherical divergence" correction but a "geometric correction" as for true spherical divergence loss the amplitude scales by a 1/z which for a constant sound-speed medium is proportional to 1/t.

line 158 - an SVD filter can be effective in removing direct wave and maybe worth trying, though extreme care is needed to get offsets correct and correctly estimate of surface mixed layer sound-speed.

Fig 3 - plot sections in the same orientation and spatially lined up so it is possible to appreciate the similarity/differences in the two images but note in caption or by arrow on section the acquisition direction.

line 183 - the subsurface will contain a range of reflection coefficients some will be tens

to hundreds times larger but others will be of the order of magnitude as SO.

lines 220-224 - definition of epochs and iterations is not clear.

General question about noise - it is not clear, or I have overlooked the statement in the paper, but was the noise section extracted from data before or after divergence correction? If so, have you not imposed a time scaling on the noise as environmental noise levels would be expected to remain constant with time? So should this denoising be applied to non-divergence corrected data?

line 290 - what is the "Static 94 synthetic seismic section?

Figs 16 & 17 see request for Fig 3.

A useful analysis would be to generate a synthetic with the expected spectral slopes then add noise at different levels and try to recover the input, the question I would like to know is is the shift after filtering (shown in Fig 19) removing weak signal too. Also discussion on the expected horizontal resolution. You state the peak frequency is 250 Hz which, after migration, should give a maximum horizontal resolution of $\sim$1.5 m. However, it will be less as this is a 2D profile over a 3D structure so there will be out-of-plane contamination.

---

## Author Comment (AC1) · 18 May 2020

Thank you for your careful review and constructive comments. We have studied all of your comments carefully and revised our manuscript. Followings are the response to the Reviewer 1's comments.

Q1. Li 6-7. The opening sentence of the abstract is a bit confusing. SO exploits water column reflections to interpret the oceanic features (fronts, eddies, water mass boundaries) as well as ocean fine structure (internal waves etc.). Futhermore, "compensating for the drawbacks of conventional PO equipment" is a very strong (and erroneous) statement. Perhaps "supplements the conventional PO observations". We should also

be very careful when interpreting the seismic images to describe PO quantitatively.

A1. We modified the sentence to "......... to supplement the conventional physical oceanographic observation methods"

Q2. Li 8. The low / high frequency band introduction is not very helpful unless you relate it to spatial resolution.

A2. We added the vertical resolution of each equipment as "Most SO studies obtain data using air guns, which have relatively low-frequency bands with vertical resolution around or larger than ten meters. For higher-frequency bands with vertical resolution ranging from several centimeters to several meters,..."

Q3. Li 10. Reword "To solve the problem"? For example, "To extract reliable signal from the low S/N . . .."

A3. We modified the sentence to "To attenuate the random noise and extract reliable signal from the low S/N ratio of sparker SO data, we applied machine learning."

Q4. Li 23: "measurements [from cruises] are performed. . .. . . observation [stations]."

A4. We modified the sentence to "Conventional physical oceanography measurements from cruises are performed by dropping equipment at the observation stations."

Q5. Li 27: mention how the sea water characteristics can be estimated (through the acoustic impedance contrasts and expand a bit more to inform the reader)

A5. We modified the sentence to "Holbrook et al. (2003) suggested a seismic oceanography (SO) method that obtained water column reflections via seismic exploration and analyzed seismic sections to estimate the oceanographic characteristics of sea water. The difference of temperature and salinity between the Labrador Current and the North Atlantic Current generated the difference of acoustic impedance which reflected the seismic signals, and the reflected signals recorded at the receivers were processed to image the thermohaline find structure of the Atlantic Ocean."

Q6, Q7. L29-32: Fine, but please do not oversell. Perhaps mention "qualitative images" and then move to "quantitative information after careful analysis where temperature/salinity contrasts produce well-defined horizons of seismic reflections" or similar. Also SO is not "widely used" Li 32: reword "determine the behavior of turbulence and internal waves" to, for example, "quantify the internal wave spectral distribution and infer turbulence"

A6., A7. We modified the sentence to "Therefore, SO is used to image the structure of water layers (Tsuji et al., 2005; Sheen et al., 2012; Piété et al., 2013; Moon et al., 2017) and provide quantitative information such as physical properties (i.e, temperature, salinity) (Papenberg et al., 2010; Blacic et al., 2016; Dagnino et al. 2016; Jun et al., 2019) or spectral distribution of the internal wave and turbulence (Sheen et al., 2009; Holbrook et al., 2013; Fortin et al. 2016) after careful analysis where temperature or salinity contrasts produce clear seismic reflections"

Q8. Li 33: clarify what central frequency is (since the source covers a range of frequencies)

A8. We modified the central frequency to "peak frequency" which is more widely used word.

Q9. Li 43: vertical resolution of 1.5 m is not much superior to the vertical resolution of "several meters" stated in line 34. Perhaps specify the latter as 5-10 m?

A9. We modified "several meters" to "around or larger than ten meters"

Q10. Li 67-68: If not using MLP and AE (and any other acronym), no need to introduce them. It is difficult to read the text.

A10. We removed the explanation of MLP and AE

Q11. Li 64-75: If there's a possibility to thin out various methods introduced (and refer to a few key references and citations therein), it can be easier for the reader to follow.

A11. We thinned and removed several explanations.

Q12. Li 77: East Sea appears very abruptly here, out of context.

A12. We removed the sentence "Therefore, this study applies the DnCNN to attenuate random noise in East Sea sparker SO data".

Q13. Li 135: delete "On the other hand,"?

A13. We removed "on the other hand"

Q14. Sec 2.1 and 2.2: can any of these descriptions refer to Fig 1? (I only see a reference in the end, at li 141, and it is not very instructive.)

A14. We relocated the sentence "Fig. 1 shows the DnCNN architecture used in this study, where Conv and BN indicate convolution and batch normalization, respectively." at the early part of the paragraph and matched explanation of each block to Fig. 1.

Q15. Li 147: This is actually one line, but two repeats (in different travelling directions). Please mention the date of data collection, vessel speed during data collection. Transect duration etc.

A15. We added date of data collection with transect duration and vessel speed.

Q16. Li 163-164: there're CTD /XCTD profiles, but the authors shown only 2xtemperature profiles from XBTs. It would be nice to increase the oceanographic context in the paper.

A16. We added 2 XCTD data.

Q17. Li 167: please describe what a reflection coefficient is.

A17. We added "Reflection coefficient is a value that defines the ratio between the reflected and incident wave."

Q18. Li 184-185: what do you mean by "thus, the subsurface seismic data have a better S/N ratio than the SO data."? Is subsurface seismic data not SO data? I suspect

you mean beneath seabed by subsurface. Please clarify.

A18. We modified the "subsurface" to "below the sea floor and beneath seabed" We also modified "sparker subsurface seismic data" to "SEZ seismic data"

Q19. Li 187: It is confusing: "We used the interval from 0.2 to 0.6 s of the original data where the noise level is relatively low". Earlier you mentioned that part was just noise!

A19. There might be misunderstanding. The interval from 0.2 to 0.6 s of the SEZ data contains seismic data below the sea floor because the SEZ data is obtained shallow part of the East Sea where the water depth is approximately shallower than 200 m. However, the East Sea SO data which is the target data of this study is obtained from the deep part of the East Sea and the water depth is approximately deeper than 1000 m. Therefore, the East Sea SO data contains random noise below 0.28 s and SEZ data contains high S/N signal between 0.2 to 0.6 s.

Q20. Li 190: Reword "the data are field data recorded with the same equipment." as "the data are collected by the same equipment"

A20. We modified the sentence.

Q21. Li 204: what is g/cc? Please use SI units.

A21. We modified g/cc to g/cm^3

Q22. Li 249: bottom right (instead of right bottom)

A22. We modified right bottom to bottom right

Q23. Li 249-250: The sentence is confusing: ". . .using training dataset 1 has one problem. The ground truth of test data 5 contains noise in the right bottom part, and training dataset 1 also contains noise in some parts of the ground truth". Dataset 1 has 6 test data. With the last reference to dataset 1 do you mean test data 1 or the entire dataset 1? Perhaps cut out the entire last part after the comma. Overall, I would appreciate a more distinct wording for test data. For example subset 1 to 6, or patch

(you use it in line 280)?

A23. We modified "test data 5" to "5th patch of the test data. We also modified the "test data" to "patch".

Q24. Li 279-280: 20th and 30th traces from the last patch: which epoch is this? Are the traces from the 50x50 patch? Can you please mention for the reader: ". . .traces out of the 50x50 size patch 6 of the test data".

A24. We modified the sentence to "We extracted the 20th (Fig. 11 (a)) and 30th (Fig. 11 (b)) vertical traces from the last (6th) patch of the test data, which is 50x50 size, as shown in Fig. 9. For the denoised trace, we extracted trace from the denoised patch of the 40th epoch".

Q25. Li 310-311: can be cut out; simply cross reference Fig 13 after 25 epochs. Overall there are repetitions throughout the authors could try to simplify.

A25. We simplified the paragraph by removing some repetitions.

Q26. Li 325: perhaps specify, "is the number of test data patch (3072)"

A26. We modified the sentence.

Q27. Li 327-328: too many significant digits at RMS errors? (perhaps enough with 6.37 and 6.34). For which epoch are these values? (Also the normalized values in line 331 could be 0.27 and 0.15)

A27. We modified 6.374 to 6.37 and 6.339 to 6.34. They are the RMS error of the test data before applying DnCNN. To clarify, we modified "RMS errors of test dataset 1 and 2 before noise attenuation. . ." to "initial RMS errors of test dataset 1 and 2 before noise attenuation. . ."

Q28. Li 332: delete "than that of the D1 model"

A28. We removed "than that of the D1 model".

Q29. Eq 6, is a division by nmode missing?

A29. We wanted to calculate the average RMS error of each test data patch (not each node in a patch). Therefore, we divided the errors by ntest only.

Q30. Li 364: "The data slope spectrum is the slope spectrum. . ." this is all very confusing. The data slope spectrum is first referred to in line 276-277 (again without explanation). Please introduce what the data slope is. For example, "the slope spectrum is the horizontal wavenumber, k_x, spectrum of the horizontal gradient of the vertical displacement of a digitized horizon. The data slope spectrum is . . .?" (or a similar explanation. Note my interpretation of the slope spectrum can be in error.)

A30. To explain the data slope spectrum and avoid confusion, we removed "data slope spectrum" in line 276-277 which is unnecessary. Instead, we modified the explanation of data slope epctrum in line 364 as "The data slope spectrum is a horizontal slope spectrum obtained directly from seismic data by calculating the horizontal wavenumber (k_x) spectrum of the seismic reflection amplitude and it is useful to identify noise contamination of seismic data and the cutoffs from an internal wave to turbulence subrange clearly (Holbrook et al., 2013; Fontin et al., 2017)."

Q31. Li 367: replace "we calculated the data slope spectrum . . .. and compared the data slope spectra" with "we calculated and compared the data slope spectra using the outcome of the D1 and D2 models. . ..

A31. We modified the sentence.

Q32. Li 376: "slope" is missing before "at wavenumbers"

A32. We added "slope".

Q33. Li 377-378: I cannot quite follow the subranges and the mentioned slopes in this panel. Perhaps mark on the figure?

A33. We marked guide lines of each subrange in figure.

Q34. Li 390: Here again mention why sparker SO data may be desirable

A34. We added "Despite the low S/N problem, the sparker source has advantage of generating relatively high frequency band signal, which can provide information with higher vertical resolution."

Q35. Fig 1. Please offer some more explanation in the caption. If not possible, defer reader to the main text

A35. We added explanation of figure in caption.

Q36. Fig 2. Elevation is grayed out for >0m, so the colorbar can stop at 0. It would be useful to add a few isobaths. I would call Line 1 and Line 2, Repeat 1 and Repeat 2.

A36. We added isobaths in the figure.

Q37. Figs 4 and 5 can be combined into 1 figure. I suggest two panels, T profiles in one panel with different color. Reflection coeff in the second panel with different colors and one offset by 0.0001 unit. Does the coefficient have a unit?

A37. We merged and modified figures. The reflection coefficient does not have a unit.

Q38. Fig 6 can be removed. It is already shown in Fig 3 and with the statement time > 0.28 s. You can mark the region by a rectangle in Fig 3.

A38. We modified the Figure.

Q39. Fig 10 (and Fig 13)- this is the average PSNR and SSIM for the 6 subsets of dataset 1? Would it not be better to show all 6 lines, or the average with one standard deviation? Actually the number of test data is 3072 (6 is an arbitrary pick), why not show the mean and std over all 3072? And also show a histogram?

A39. It is the average PSNR and SSIM of dataset 1 (Figure 10) and dataset 2 (Figure 13) We calculated standard deviation over all 3072 test data.

Q40. Fig 11 caption: in the end the cross reference should be to Fig 9.
A40. We modified the cross reference.

Q41. Fig 14 caption: in the end the cross reference should be to Fig 12. In the text the model is referred to as D2 (but here D1).

A41. We modified the cross reference.

Q42. Fig 16. This figure is not needed either. It is simply the upper 0.28s of Fig 3. However, I appreciate that it is zoomed in and compared to the cleaned sections. See suggestion below Fig 17 comment. Fig 17. Please consider removing xaxis labels from panels c to d, and placing panel labels in the upper left corner of panels (it's grayed out anyway), so that you can have a more condensed 4-panel, 1 page figure with minimum white space vertically between panels. I think a reorganized version of Figs 16- 18 will be much better for the reader. I suggest 2 figures, each with 5 panels (with identical x-axis limits and width, and minimum white space between them). New Fig 16: Line 1 results. New Fig 17 Line 2 results with corresponding 5 panels for each figure: a) data (as in Fig 16a) b) clean after D1 c) clean after D2 d) noise after D1 e) noise after D2.

A42. We re-ordered the figures and made new Figure 16, 17. The layout of the figures will be re-arrayed at the publication stage. We will request the Journal to place the Figure 16 and 17 at the same page because it would be better for reader to read. If it is difficult, we will modify the figure to make it possible.

* We are still modifying the manuscript and will try to improve the manuscript more.

* There may be some changes in the sentences which are not grammatically correct in the future because we will get the English editing service after finishing revise.

---

## Author Response (AR1)

**Reviewr 1**

Thank you for your careful review and constructive comments. We have studied all of your comments carefully and revised our manuscript. Followings are the response to the Reviewer 1's comments. "Q" is a comment from the reviewer, "A" is a response to the comment and "Changes" the lines and details of the modification.

**Q1.** Li 6-7. The opening sentence of the abstract is a bit confusing. SO exploits water column reflections to interpret the oceanic features (fronts, eddies, water mass boundaries) as well as ocean fine structure (internal waves etc.). Futhermore, "compensating for the drawbacks of conventional PO equipment" is a very strong (and erroneous) statement. Perhaps "supplements the conventional PO observations". We should also be very careful when interpreting the seismic images to describe PO quantitatively.

A1. We modified the sentence.

**Changes: Line 6-7.** We modified the opening sentence to "Seismic oceanography (SO) acquires water column reflections using controlled source seismology and provides high lateral resolution that enables the tracking of the thermohaline structure of the oceans."

Q2. Li 8. The low / high frequency band introduction is not very helpful unless you relate it

to spatial resolution.

A2. We added the approximate frequency range with vertical resolution of each equipment.

**Changes: Line 10-11.** We added the resolution as "with a vertical resolution of approximately ten meters or more" and "with vertical resolution ranging from several centimeters to several meters."

Q3. Li 10. Reword "To solve the problem"? For example, "To extract reliable signal from the low S/N . . .."

**A3.** We modified the sentence.

Changes: Line 14. We modified the sentence as "To attenuate the random noise and extract reliable signal from."

Q4. Li 23: "measurements [from cruises] are performed. .... observation [stations]."

A4. We modified the sentence.

**Changes: Line 28.** The modified sentence is "Conventional physical oceanography measurements from cruises are performed by dropping equipment at the observation stations"

**Q5.** Li 27: mention how the sea water characteristics can be estimated (through the acoustic impedance contrasts and expand a bit more to inform the reader)

A5. We added the explanation how the sea water is imaged

**Changes: Line 32-34.** The added sentence is "The differences in temperature and salinity between water column generate the difference in acoustic impedance, which reflect the seismic signals, and the reflected signals recorded at the receivers are processed to image the thermohaline fine structure of the ocean."

Q6, Q7. L29-32: Fine, but please do not oversell. Perhaps mention "qualitative images" and then move to "quantitative information after careful analysis where temperature/salinity contrasts produce well-defined

horizons of seismic reflections" or similar. Also SO is not "widely used"

Li 32: reword "determine the behavior of turbulence and internal waves" to, for example, "quantify the internal wave spectral distribution and infer turbulence"

A6., A7. We modified the sentence.

**Changes: Line 37-41.** The modified sentence is "Therefore, SO is used to image the structure of water layers (Tsuji et al., 2005; Sheen et al., 2012; Piété et al., 2013; Moon et al., 2017) and provide quantitative information such as physical properties (i.e, temperature, salinity) (Papenberg et al., 2010; Blacic et al., 2016; Dagnino et al. 2016; Jun et al., 2019) or the spectral distribution of the internal wave and turbulence (Sheen et al., 2009; Holbrook et al., 2013; Fortin et al. 2016) after careful analysis where temperature or salinity contrasts produce clear seismic reflections"

Q8. Li 33: clarify what central frequency is (since the source covers a range of frequencies)

A8. We added the definition of the central frequency.

Changes: Line 44. We added "the geometric center of the frequency band (Wang, 2015)."

**Q9.** Li 43: vertical resolution of 1.5 m is not much superior to the vertical resolution of "several meters" stated in line 34. Perhaps specify the latter as 5-10 m?

**A9.** We modified specify the resolution.

Changes: Line 46. We modified "several meters" to "approximately ten meters or more."

Q10. Li 67-68: If not using MLP and AE (and any other acronym), no need to introduce them. It is difficult to read the text.

A10. We removed the explanation of MLP and AE.

**Changes: Line 81-82.** We removed the sentence "Noise attenuation using machine learning has been widely studied, such as the multilayer perceptron (MLP) (Burger et al., 2012) and autoencoder (AE) (Xie et al., 2012; Wu et al., 2016)."

**Q11.** Li 64-75: If there's a possibility to thin out various methods introduced (and refer to a few key references and citations therein), it can be easier for the reader to follow.

A11. We thinned and removed several explanations.

**Changes: Line 77-91.** We rewrite this part as "The use of artificial intelligence (AI) has been studied in geophysics for decades (McComack, 1991; McCormack et al., 1993; Van der Baan and Jutten, 2000), but recent advances in computer resources and algorithms have spurred AI research, and several studies have been conducted to apply machine learning in the field of seismic data processing (Araya-Polo et al., 2019; Yang and Ma, 2019; Zhao et al., 2019). Among them, one of the most actively studied areas is prestack and poststack data noise attenuation. After convolutional neural networks (CNNs) were introduced, various noise attenuation methods based on the CNN architecture have been proposed (Jian and Seung, 2009; Gordonara, 2016; Lefkimmiatis, 2017), and the denoising convolutional neural network (DnCNN) suggested by Zhang et al. (2017) attained good results in random noise suppression in natural images. Recently, the DnCNN was applied to attenuate various types of noise in seismic data (Li et al., 2018; Si and Yuan, 2018; Liu et al., 2018). The DnCNN uses residual learning (He et al., 2016) and has the advantage of minimizing damage to the seismic signal by estimating the noise from seismic data rather than directly analyzing the signal. The original shape of the water column reflector in SO data remains unchanged during data processing, so the DnCNN, which learns noise characteristics, is a suitable SO data denoising

algorithm."

Q12. Li 77: East Sea appears very abruptly here, out of context.

A12. We removed the sentence of "East Sea".

Changes: Line 91-92. We removed the sentence "Therefore, this study applies the DnCNN to attenuate random noise in East Sea sparker SO data."

Q13. Li 135: delete "On the other hand,"?

A13. We removed "on the other hand"

**Changes: Line 153.** The modified sentence is "This study extracts noise from binary files, and a  $3 \times 3 \times 1$  convolution filter is adopted."

Q14. Sec 2.1 and 2.2: can any of these descriptions refer to Fig 1? (I only see a reference in the end, at li 141, and it is not very instructive.)

A14. We relocated the sentence

**Changes: Line 145-148.** The sentence "Fig. 1 shows the DnCNN architecture used in this study, where Conv and BN indicate convolution and batch normalization, respectively." is now located at the early part of the paragraph and matched explanation of each block to Fig. 1 like "This layer is shown as "Conv+ReLU" in Fig. 1."

**Q15.** Li 147: This is actually one line, but two repeats (in different travelling directions). Please mention the date of data collection, vessel speed during data collection. Transect duration etc.

A15. We added date of data collection with transect duration and vessel speed.

**Changes: Line 167-169.** The sentence "The survey was performed from October 7th to 11th in 2018 (approximately 38 hours for one line) and the vessel speed was 5.5 knots." is added.

**Q16.** Li 163-164: there're CTD /XCTD profiles, but the authors shown only 2xtemperature profiles from XBTs. It would be nice to increase the oceanographic context in the paper.

A16. We added 2 XCTD data.

**Changes: Line 185.** Temperature and reflection coefficients information from Two XCTDs are added in Fig. 4(a) and (b).

Q17. Li 167: please describe what a reflection coefficient is.

A17. We explained what a reflection coefficient is.

**Changes: Line 187-188.** The modified sentence is "Fig. 4 (b) shows the reflection coefficients, defining the ratio between the reflected and incident wave, calculated with the XBT and XCTD data."

**Q18.** Li 184-185: what do you mean by "thus, the subsurface seismic data have a better S/N ratio than the SO data."? Is subsurface seismic data not SO data? I suspect you mean beneath seabed by subsurface. Please clarify.

A18. We clarified the meaning.

**Changes: Line 205, Line 212.** We modified the "subsurface" to "below the sea floor and beneath seabed." We also modified "sparker subsurface seismic data" to "SEZ seismic data"

**Q19.** Li 187: It is confusing: "We used the interval from 0.2 to 0.6 s of the original data where the noise level is relatively low". Earlier you mentioned that part was just noise!

A19. There might be misunderstanding. The interval from 0.2 to 0.6 s of the SEZ data contains seismic data below the sea floor because the SEZ data is obtained shallow part of the East Sea where the water depth is approximately shallower than 200 m. However, the East Sea SO data which is the target data of this study is obtained from the deeper part of the East Sea and the water depth is approximately deeper than 1000 m. Therefore, the East Sea SO data contains random noise below 0.28 s (the water column) and SEZ data contains high S/N signal between 0.2 to 0.6 s (beneath the sea bed).

**Q20.** Li 190: Reword "the data are field data recorded with the same equipment." as "the data are collected by the same equipment"

A20. We modified the sentence.

**Changes: Line 213-215.** The modified sentence is "This method has the advantage of using data with similar characteristics to those of the target data (the East Sea SO data) as the ground truth because the data are collected by the same equipment."

Q21. Li 204: what is g/cc? Please use SI units.

A21. We changed unit to SI unit.

Changes: Line 228. We changed "g/cc" to "1 g/cm3"

Q22. Li 249: bottom right (instead of right bottom)

A22. We modified "right bottom" to "bottom right"

Changes: Line 275.

**Q23.** Li 249-250: The sentence is confusing: ". . . using training dataset 1 has one problem. The ground truth of test data 5 contains noise in the right bottom part, and training dataset 1 also contains noise in some parts of the ground truth". Dataset 1 has 6 test data. With the last reference to dataset 1 do you mean test data 1 or the entire dataset 1? Perhaps cut out the entire last part after the comma. Overall, I would appreciate a more distinct wording for test data. For example, subset 1 to 6, or patch (you use it in line 280)?

A23. We clarified the sentences.

**Changes: Line 264-275.** We modified "ground truth of test data 5" to "ground truth of the 5th test data patch". We also modified the "test data" to "test data patch".

**Q24.** Li 279-280: 20th and 30th traces from the last patch: which epoch is this? Are the traces from the 50x50 patch? Can you please mention for the reader: ". . .traces out of the 50x50 size patch 6 of the test data".

A24. We modified the sentence.

**Changes:** Line 308-310. The modified sentence is "We extracted the  $20^{\text{th}}$  (Fig. 10 (a)) and  $30^{\text{th}}$  (Fig. 10 (b)) vertical traces from the last (6th) patch of the test data, which had a size of 50x50. For the denoised trace, we extracted trace from the denoised patch of the  $40^{\text{th}}$  epoch."

**Q25.** Li 310-311: can be cut out; simply cross reference Fig 13 after 25 epochs. Overall there are repetitions throughout the authors could try to simplify.

A25. We simplified the paragraph by removing some repetitions.

**Changes: Line 342-343.** The sentence "Similar to the previous experiment, we also calculated the average PSNR and SSIM to quantitatively verify the test results and compared the amplitudes of the extracted traces. Fig. 13 (a) shows the average PSNR, and 13(b) shows the average SSIM." was removed.

Q26. Li 325: perhaps specify, "is the number of test data patch (3072)"

**A26.** We modified the sentence.

Changes: Line 359. The sentence is modified as "is the number of test data patches (3,072)"

**Q27.** Li 327-328: too many significant digits at RMS errors? (perhaps enough with 6.37 and 6.34). For which epoch are these values? (Also the normalized values in line 331 could be 0.27 and 0.15)

A27-1. We reduced the significant digits.

**A27-2.** They are the RMS error of the test data before applying DnCNN. We modified the sentence to clarify the meaning.

**Changes: Line 361-365.** To clarify, we modified "RMS errors of test dataset 1 and 2 before noise attenuation..." to "initial RMS errors of test dataset 1 and 2 before noise attenuation..." We also changed 6.374, 6.339, 0.268, 0.151 to 6.37, 6.34, 0.27, 0.15.

Q28. Li 332: delete "than that of the D1 model"

A28. We removed "than that of the D1 model".

Changes: Line 366-367.

Q29. Eq 6, is a division by nmode missing?

**A29.** We wanted to calculate the average RMS error of each test data patch (not each node in a patch). Therefore, we divided the errors by ntest only.

**Q30.** Li 364: "The data slope spectrum is the slope spectrum..." this is all very confusing. The data slope spectrum is first referred to in line 276-277 (again without explanation). Please introduce what the data slope is. For example, "the slope spectrum is the horizontal wavenumber,  $k_x$ , spectrum of the horizontal gradient of the vertical displacement of a digitized horizon. The data slope spectrum is . . .?" (or a similar explanation. Note my interpretation of the slope spectrum can be in error.)

**A30.** To explain the data slope spectrum and avoid confusion we, we removed "data slope spectrum" in line 276-277 (in the original manuscript) which is unnecessary. Instead, we added the explanation of data slope spectrum at the synthetic data slope spectrum experiment part.

**Changes: Line 305-306 and Line 368-374.** We removed "In particular, the amplitude information is a key parameter for acquiring the data slope spectrum, which calculates slope spectra directly from the seismic data (Holbrook et al., 2013; Fortin et al., 2017)." and added "Water column reflection data can be used to obtain the physical oceanographic information by calculating the slope spectrum. The data slope spectrum is a horizontal slope spectrum obtained directly from seismic data by calculating the horizontal wavenumber () spectrum of the seismic reflection amplitude, and it is useful to identify noise contamination of seismic data and the cutoffs from an internal wave to turbulence subrange (Holbrook et al., 2013; Fontin et al., 2017). Holbrook et al. (2013) suggested calculating the data slope spectrum before calculating the reflector slope spectrum because the random noise that should be removed before analyzing the seismic data becomes evident in the data slope spectrum."

Q31. Li 367: replace "we calculated the data slope spectrum . . . . and compared the data slope spectra" with "we calculated and compared the data slope spectra using the outcome of the D1 and D2 models. . ..

A31. We modified the sentence.

**Changes: Line 444-445.** We modified the sentence to "To validate the noise attenuation results, we also calculated and compared the data slope spectra by using the outcome of the D1 and D2 models."

Q32. Li 376: "slope" is missing before "at wavenumbers"

A32. We added "slope".

Changes: Line 454.

**Q33.** Li 377-378: I cannot quite follow the subranges and the mentioned slopes in this panel. Perhaps mark on the figure?

A33. We marked guide lines of each subrange in figure.

Changes: Line 457-458 and Fig. 20.

Q34. Li 390: Here again mention why sparker SO data may be desirable

A34. We mentioned why sparker SO data is desirable again.

**Changes: Line 474-475.** "Despite the low S/N problem, the sparker source has advantage of generating relatively high frequency band signal, which can provide information with higher vertical resolution."

Q35. Fig 1. Please offer some more explanation in the caption. If not possible, defer reader to the main text

A35. We added explanation of figure in caption of Fig. 1.

Changes: Fig. 1.

**Q36.** Fig 2. Elevation is grayed out for >0m, so the colorbar can stop at 0. It would be useful to add a few isobaths. I would call Line 1 and Line 2, Repeat 1 and Repeat 2.

A36. We added isobaths in the Fig. 2.

Changes: Fig. 2.

**Q37.** Figs 4 and 5 can be combined into 1 figure. I suggest two panels, T profiles in one panel with different color. Reflection coeff in the second panel with different colors and one offset by 0.0001 unit. Does the coefficient have a unit?

A37. We merged and modified figures, and the reflection coefficient does not have a unit.

Changes: Fig4.

**Q38.** Fig 6 can be removed. It is already shown in Fig 3 and with the statement time > 0.28 s. You can mark the region by a rectangle in Fig 3.

A38. We modified the Figure and marked the noise part by using red box.

Changes: Fig. 3.

**Q39.** Fig 10 (and Fig 13)- this is the average PSNR and SSIM for the 6 subsets of dataset 1? Would it not be better to show all 6 lines, or the average with one standard deviation? Actually the number of test data is 3072 (6 is an arbitrary pick), why not show the mean and std over all 3072? And also show a histogram?

A39-1. It is the average PSNR and SSIM of dataset 1 (Fig. 8) and dataset 2 (Fig. 12)

A39-2. We calculated standard deviation over all 3,072 test data and plotted.

Changes: Fig. 8 and 12.

Q40. Fig 11 caption: in the end the cross reference should be to Fig 9.

A40. We modified the cross reference.

Changes: Fig. 10.

**Q41.** Fig 14 caption: in the end the cross reference should be to Fig 12. In the text the model is referred to as D2 (but here D1).

A41. We modified the cross reference.

Changes: Fig. 14.

**Q42.** Fig 16. This figure is not needed either. It is simply the upper 0.28s of Fig 3. However, I appreciate that it is zoomed in and compared to the cleaned sections. See suggestion below Fig 17 comment.

Fig 17. Please consider removing xaxis labels from panels c to d, and placing panel labels in the upper left corner of panels (it's grayed out anyway), so that you can have a more condensed 4-panel, 1 page figure with minimum white space vertically between panels.

I think a reorganized version of Figs 16- 18 will be much better for the reader. I suggest 2 figures, each with 5 panels (with identical x-axis limits and width, and minimum white space between them). New Fig 16: Line 1 results. New Fig 17 Line 2 results with corresponding 5 panels for each figure:

a) data (as in Fig 16a) b) clean after D1 c) clean after D2 d) noise after D1 e) noise after D2.

A42. We re-ordered the figures. We also merged figures by removing unnecessary labels and white spaces.

**Changes: Fig. 18 and 19.**

**General comment:**

I would strongly encourage the authors to improve the "Ocean Science" part by adding more insight using the clean seismic sections. Can you make some oceanographic inferences, interpretations (or better quantification) using the cleaned Lines 1 and 2?

→ This paper tries to apply machine learning technology to remove random noise from sparker SO data to help interpret SO data, and confirm the possibility of quantitative analysis. Because we want to focus on the noise attenuation method itself, the machine learning methodology and application of the proposed method are the main part of the paper. Adding the oceanographic analysis of denoised data will help authors understand the characteristic of the East Sea, but the manuscript will be vast in content. Therefore, after confirming the possibility of oceanographic analysis using denoised sparker SO data in this study, the detailed oceanographic analysis of East Sea data will be performed in the future study.

Finally, it would serve the community much better if the authors made available some code for noise attenuation using machine learning. They offer the code through communication with authors, but the impact would be far larger if they make the code available as a supplement.

→ We also agree your comment. We think the distribution of the code is necessary for the community. However, the program will undergo some modification because the review process is not finished. After finishing the review process, we will distribute the program through github.

**Reviewr 2**

Thank you for your careful review and constructive comments. We have studied all of your comments carefully and revised our manuscript. We edited the English of the entire manuscript including "Abstract" by following Reviewer 2's recommendation.

This paper deals with the noise attenuation method of sparker SO data using machine learning. The data obtained from the sparker source have advantages such as cheap data acquisition costs and high vertical resolution from several centimeters to several meters, but it has not been widely used in SO study and has not been quantitatively analyzed to date. This is mainly because of the low S/N ratio of the sparker seismic data. Due to strong noise, the conventional data processing method is not sufficient to attenuate the noise in the sparker seismic data, thus it is difficult to perform quantitative analysis such as calculating slope spectrum. Therefore, we would like to propose a method to suppress random noise in the sparker seismic data. This paper tries to apply machine learning technology to remove random noise from sparker SO data to help interpret SO data, and confirm the possibility of quantitative analysis. Because of this reason, the machine learning methodology and application of the proposed method are the main part of the paper. After confirming the possibility of oceanographic analysis using denoised sparker SO data in this study, the detailed oceanographic analysis of East Sea data will be performed in the future study. Followings are the response to the Reviewer 2's comments. "Q" is a comment from the reviewer, "A" is a response to the comment and "Changes" the lines and details of the modification.

**Q1.** line 34 - delete "relatively low" as you do not state relative to what. Please edit paper to remove, as much as is possible, unqualified comparative statements.

A1. We removed "relatively low" in line 34 and removed unqualified comparative statements in the manuscript.

Changes: Line 9, 45, 52, 61, 211, 277 and 473.

**Q2.** line 46 - This problem is more accentuated in SO because the impedance contrasts between the layers are small.

**A2.** We modified the sentence.

**Changes: Line 57-58.** The modified sentence is "This problem is more accentuated in SO because the impedance contrasts between the water layers are smaller than the impedance contrasts between the layers beneath the seabed."

**Q3.** line 65 - this reference list ignores the long history of the use of Neural Networks see McCormack's paper in Leading Edge 1991 which shows an early attempt to use these NN to identify noisy traces in seismic data, since then NN have been evaluated for many tasks in the processing of seismic reflection data. Suggest authors change sense to recognise the history but equally highlight the recent advances in AI. I now note that this history is partly addressed in the following paragraph.

A3. We modified the sentence and added the history of Neural Network in seismic data processing.

**Changes: Line 77-80.** The modified sentence is "The use of artificial intelligence (AI) has been studied in geophysics for decades (McComack, 1991; McCormack et al., 1993; Van der Baan and Jutten, 2000), but recent advances in computer resources and algorithms have spurred AI research, and several studies have been conducted to apply machine learning in the field of seismic data processing (Araya-Polo et al., 2019; Yang and Ma, 2019; Zhao et al., 2019)."

Q4. line 153 - scaling by the sq-rt of time is not "spherical divergence" correction but a "geometric correction" as for true spherical divergence loss the amplitude scales by a 1/z which for a constant sound-speed medium is proportional to 1/t.

A4. We scaled the data by the sq-rt of time because we tried to make balance between noise in the shallow part which is affected by the tails of complex source wavelet and in the deep part of the data. We modified the phrase.

Changes: Line 173-174. We modified "spherical divergence correction" to "amplitude correction."

Q5. line 158 - an SVD filter can be effective in removing direct wave and maybe worth trying, though extreme care is needed to get offsets correct and correctly estimate of surface mixed layer sound-speed

**A5.** To remove the direct wave, SVD filter or Tau-P domain filter would be appropriate. However, the source signature of the sparker data is more complex than that of the air gun data, thus the filter may not properly eliminate the direct wave. Moreover, the noise near the sea surface is severe and the section before 0.03 second is not our research target (interesting SO signal does not exist in that part because this part is mixed layer which does not have large differences in reflection coefficient), therefore we muted the section before 0.03 second.

**Q6.** Fig 3 - plot sections in the same orientation and spatially lined up so it is possible to appreciate the similarity/differences in the two images but note in caption or by arrow on section the acquisition direction.

A6. We plotted the sections in the same orientation and added an arrow indicating the ship direction.

**Changes: Fig 3.**

**Q7.** line 183 - the subsurface will contain a range of reflection coefficients some will be tens to hundreds times larger but others will be of the order of magnitude as SO.

**A7.** We modified the sentence.

**Changes: Line 205-207.** The modified sentence is "The reflection coefficients of the major reflectors below the sea floor are tens to hundreds of times larger than that of the water column; thus, the seismic data below the sea floor have a better S/N ratio than the SO data."

Q8. lines 220-224 - definition of epochs and iterations is not clear.

**A8.** We modified the sentence to clarify the definition of epoch and iteration.

**Changes: Line 247.** The modified sentence is "The epoch is a process using all training data, and iteration is a process using a mini-batch; thus, an epoch usually consists of several iterations."

**Q9.** General question about noise - it is not clear, or I have overlooked the statement in the paper, but was the noise section extracted from data before or after divergence correction? If so, have you not imposed a time scaling on the noise as environmental noise levels would be expected to remain constant with time? So should this denoising be applied to non-divergence corrected data?

**A9-1.** We extracted the noise from the processed seismic section which was applied the amplitude correction. Even though the background noise level is supposed to be not influenced by the time, the noise level at the early time in the East Sea SO data is larger than the deep part of the section (this might be the noise related to the complex source wavelet of sparker). Therefore, we empirically selected square root of time as scaling factor to make balance of the noise amplitude from shallow to deep part of the section.

**A9-2.** Since we extracted the noise from the amplitude corrected seismic section, we applied the trained model to the amplitude corrected seismic section to remove the random noise. If we extracted the noise from non-amplitude-corrected data, then we should apply the trained model to the non-amplitude-corrected data.

**A9-3.** Before calculating the data slope spectra, we scaled the seismic section again by multiplying square root of time to each time step (consequently multiplying time to each time step of the data) for the spherical divergence correction.

**Changes: Line 446-447.** The modified sentence is "Before calculating the data slope spectrum, we scaled the seismic sections again by multiplying the square root of time to each time step (consequently multiplying the time to each time step) for the spherical divergence correction."

Q10. line 290 - what is the "Static 94 synthetic seismic section?

A10. We modified the sentence and added the reference.

Changes: Line 320. The sentence is changed to "part of the 1994 Amoco static test dataset (SEG Wiki)" and reference is "SEG Wiki: https://wiki.seg.org/wiki/1994\_BP\_statics\_benchmark\_model, last access: 22 June 2020."

Q11. Figs 16 & 17 see request for Fig 3.

A11. We modified the Figures.

Changes: Fig. 18 and 19.

**Q12.** A useful analysis would be to generate a synthetic with the expected spectral slopes then add noise at different levels and try to recover the input, the question I would like to know is is the shift after filtering (shown in Fig 19) removing weak signal too. Also discussion on the expected horizontal resolution. You state the peak frequency is 250 Hz which, after migration, should give a maximum horizontal resolution of  $\sim 1.5$  m. However, it will be less as this is a 2D profile over a 3D structure so there will be out-of-plane contamination.

A12-1. We performed experiment using synthetic data.

Changes: Line 368-386 and Fig16. and 17. We explained the reason of performing the synthetic data experiment and showed the result of the experiment.

**A12-2.** We also can find the shifting of the spectrum between the noise added synthetic section and noise attenuated seismic sections at the wavenumber smaller than 0.001 cpm. However, the difference is also observed between the spectrum of noise free section and noise added section. In addition, the shifting is not observed between the spectrum of noise free section and noise attenuated section. Therefore, this shifting seems to be caused by the characteristic of the noise extracted from the East Sea SO data.

A12-3. We also mentioned the shift issue in the manuscript.

**Changes: Line 459-464.** The explanation of shift issue is "There is a shift in the data slope spectrum after noise attenuation at wavenumbers smaller than 0.001 cpm. This shift is also observed in the synthetic data slope spectrum experiments. In Fig. 17, there is a difference between the spectrum of the noise-added section and that of the noise-attenuated sections at wavenumbers smaller than 0.001 cpm. However, the difference is also observed between the spectrum of the noise-free section and that of the noise-added section. Therefore, this shift seems to be caused by the characteristic of the noise extracted from the East Sea SO data."

**A12-4.** In the conclusion, we added the limitation of 2D exploration related to the resolution. And we mentioned that it is necessary to acquire data by using 3D seismic exploration to improve the resolution

**Changes: Line 489-496.** The added sentence is "Even though the random noise is almost completely attenuated in the seismic section, the proposed method still needs several improvements. The observed random noise is successfully attenuated in the seismic section, but the data slope spectrum still indicates that the section contains noise with a slope of  $k_x^2$  at wavenumbers above 0.02 cpm. Therefore, future studies should include a detailed analysis of the slope spectra of the East Sea SO data and establish an improved noise attenuation algorithm suitable

for higher wavenumbers. Moreover, the data were collected using 2D seismic exploration, which can degrade the seismic resolution during the data processing stage because of the limitations of 2D seismic exploration such as out-of-plane contamination. Therefore, to improve the resolution of SO data, it is necessary to acquire data by using 3D seismic exploration."

**Random Noise Attenuation of Sparker Seismic Oceanography Data** with Machine Learning**

Hyunggu Jun1, Hyeong-Tae Jou1, Chung-Ho Kim1, Sang Hoon Lee1, Han-Joon Kim1

1Korea Institute of Ocean Science & Technology, Busan, 49111, Republic of Korea

5 Correspondence to: Hyeong-Tae Jou (htjou@kiost.ac.kr)

Abstract. Seismic oceanography (SO) acquires water column reflections using controlled source seismology and provides high lateral resolution that enables the tracking of the thermohaline structure of the oceans. by seismic exploration compensating for the drawbacks of conventional physical oceanographic equipment. Most SO studies obtain data using air guns, which can produce acoustic energy below 100 Hz bandwidth.have relatively low frequency bands with a vertical

- 10 resolution of approximately ten meters or more. For higher-frequency bands, with vertical resolution ranging from several centimeters to several meters, at a low exploration cost, using a smaller, low-cost seismic exploration system may be used, such as a sparker source with central frequencies of 250 Hz or higher. a shorter receiver length, would be an alternative. However, the sparker source has a relatively low energy compared to air guns and consequently produces data with a lower signal-to-noise (S/N) ratio. To solve the problem To attenuate the random noise and extract reliable signal from of the low S/N
- 15 ratio of sparker SO data without distorting the true shape and amplitude of water column reflections, we applied machine learning. The purpose of this study is to attenuate the random noise in the East Sea sparker SO data without distorting the true shape and amplitude of water column reflections. Specifically, we used A-a denoising convolutional neural network (DnCNN) that successfully suppresses random noise in a natural image. is adopted as the machine learning network architecture. One of the most important factors of machine learning is the generation of an appropriate training dataset. We have generated two
- 20 different training datasets using synthetic and field data. Models trained with the different training datasets are-were\_applied to the test data, and the denoised results are-were\_quantitatively compared. To demonstrate the technique, the The-trained models are-were\_applied to an SO sparker seismic dataset acquired in the East Sea the target seismic data, i.e., the East Sea sparker water column seismic reflection data, and the denoised seismic sections are-were\_evaluated. The results show that machine learning can successfully attenuate the random noise of in sparker water column seismic reflection data.

**1** Introduction**

[revised manuscript text omitted]

To evaluate the test result quantitatively, we calculated the peak S/N ratio (PSNR) and structural similarity index measure (SSIM) by using entire test data. The PSNR reflects the amount of noise contained in the data and can be calculated as follows (Hore and Ziou, 2010):

285

$$PSNR = 20 \log_{10}(MAX_I) - 10 \log_{10}(MSE)$$
(4)

where  $MAX_I$  is the maximum value of the image and MSE is the mean squared error between the data with and without noise. The PSNR is high when noise is successfully removed, while the PSNR is low when noise is not sufficiently removed. Fig. <del>10</del> 290 8 (a) shows the average PSNR and standard deviation of the test results. At the early stage of training, the average PSNR is low, which indicates that noise has not been sufficiently removed, but it increases as training progresses and converges at approximately 36 dB after 25 epochs. Even though the denoising algorithm attenuates noise successfully, the reflection shape, which is important information of the SO data, can be altered. Therefore, it is necessary to measure the structural distortion to verify the effectiveness of the proposed method. The SSIM is a quality metric that calculates the structural similarity between 295 two datasets and can be calculated as follows (Hore and Ziou, 2010):

[revised manuscript text omitted]

noise of sparker SO data. Therefore, we calculated the root-mean-square (RMS) error between the denoised test data and ground truth of the test data and evaluated which training data produced a lower RMS error. The RMS error was calculated as follows:

$$RMS \ error = \sqrt{\frac{1}{ntest} \sum_{i=1}^{ntest} \sum_{j=1}^{nnode} (g_{ij} - d_{ij})^2}$$
(6)

[revised manuscript text omitted]

---

## Referee Report (RR1)

Title: Random Noise Attenuation of Sparker Seismic Oceanography Data with Machine Learning

authors: Hyunggu Jun, Hyeong-Tae Jou, Chung-Ho Kim, Sang Hoon Lee, Han-Joon Kim

submission number: os-2020-13

reviewer: Richard Hobbs

General comments— The paper has an improved balance of content, though I still think much of the detail of the filtering method could be moved to supplementary material as it will be of low interest to the principal readership of this journal.

Editor to advise if American spelling is acceptable and some phasing is still awkward in places but it does not detract from the understanding.

Comments and corrections.

line 8 punctuation

line 23: "dropping equipment" ? Rephrase as this is ambiguous

line 25: first published use of technique was by Gonella, J., Michon, D., 1988. Deep internal waves measured by seismic-reflection within the eastern Atlantic water mass. Comptes Rendus de l Academie Des Sciences Serie II 306, 781–787 (in French with English abstract). However it was Holbrook who first used the term seismic oceanography.

line 27/28: "The differences in temperature and salinity between water column generate ..." poor english

line 29: need reference here suggest Ruddick, B., Song, H., Dong, C., Pinheiro, L., 2009. Water column seismic images as maps of temperature gradient. Oceanography 22 (1), 192–205.

line 31: need reference for "conventional oceanographic methods" and link to the equipment that was dropped in line 23.

line 34: redundant work "careful"

line 39: "...higher exploration expenses..." ambiguous as higher than what

line 45: the vertical resolution of 1.5 m is based on the theoretical Rayleigh limit however in a 3D environment it is unlikely that you can achieve this due to interference from out-of-plane scattering.

Line 49/50: the maximum impedance contrasts are smaller

line 53 & 64: most noise sources are not random so the authors need to define which noise sources that consider as problematical, especially for SO data where a dominant noise is caused by pressure variations at the receiver due to waves on the surface. This peaks at frequencies below ~5 Hz so with sparker SO you should do better then airguns as you can set your low-cut filter at a higher frequency without damaging your primary signal bandwidth. I think what the authors are looking at here is noise generated from the ship as it has a distinct time characteristic and I would not be

surprised if this was during periods of tidal flow against the direction of travel. So perhaps the ships propellers were run at a faster rate to maintain speed over the ground which has resulted in more cavitation. However, that period does not match with the statement that the ship speed was 5.5 knots. Suggest authors include auto-correlation functions both in space and time of the red boxes (Fig 3) indicated to justify that the synthetic random noise used in training datasets is appropriate.

Figure 2 – does this show the whole line or just the portions after the shelf sections were omitted?

Line 163 no comma in velocity value and possibly need to correct style to m s-1 and other units as necessary. Also this is ambiguous as 'water velocity' in the context of oceanography describes the mass movement of a body of water. An acceptable term is 'sound-speed'. As the temperature and salinity of the water is known (XCTD) then authors could do better than just use a generic sound-speed by using the equations of state.

Line 173; define what you mean by very small as it looks like the calculated reflection coefficients are at the level of the minimum resolution of the XBT and given that you have two XCTD why did you use a constant density for those?

Line 209: justify why you used 1000 kg m-3 for the density as true density is provided as part of the model – why does this change make the synthetic more SO-like if you don't apply a scaling an shift to the compressional wave velocities too. It sounds like a fix to overcome inefficiencies in your modelling process eg calling depth time and followed by a convolution. Given the level of adjustment and modification (see line 224), I see no purpose for using these sophisticated models as a set of time-shifted spikes (eg an enlarged version of Fig 16) would have probably done equally as well.

Line 210: were the simulations acoustic or elastic?

Line 247-260: you don't actually know the 'ground truth' as the section without added noise is, as you correctly point out, still contaminated with noise and will contain inter-layer peg-legs and outof-plane events which you algorithm will also treat as primary signal.

Line 275: what is x and y in equation 5

line 339 suggest a sub heading here to indicate you have changed to dealing with the SO data

line 342 "and" → "to reveal"

lines 342-344 suggest: Holbrook et al. (2013) suggested analysing the slope of the spectrum for the complete data before calculating the slope of the spectrum from the water reflections because the noise that needs to be suppressed is more evident in the spectrum of the complete data.

```
line 369 "clearly imaged" → "improved"
```

Line 375-378, 385-387 & 389-393: I think you are making too much of this as the differences between the approaches is marginal as shown by your fig 20. Essentially both methods worked on the data to a sufficient degree to enable meaningful estimates of the slope of the internal wave to be discerned. If you feel you want to emphasise the benefit of D2 vs D1 then show a difference plot of the two.

Line 397: what sound speed model did you use for conversion?

Line 406: would be useful to quote horizontal limit of resolution due to the Fresnel zone which I estimate, by eye-balling your data, to be ~0.02 cpm so above this value you would expect the amplitude to the perturbations to be strongly filtered so the fact the signal becomes noise limited is to be expected and further noise reduction will not provide any useful information. I note that your data appear to have a lower frequency content than I would expect – how do you explain that? What were the acquisition parameters? As these directly effect the data signal bandwidth.

---

## Author Response (AR2)

Thank you for your careful review and constructive comments. We have studied all of your comments carefully and revised our manuscript. Followings are the response to the Editor, Referee 1 and Referee 2's comments. "Q" is a comment from the reviewer, "A" is a response to the comment and "Change" the lines and details of the modification. The comments are black in color and responses / changes are blue in color.

**Editor's comments.**

Q1. Line 67. "McCormack" (spelling) A1. We modified the typo. Change: Line 73.

Q2. Lines 95-98. The text and figure 1 do not match well here despite the figure being cited: no "residual learning" in figure 1; no ReLU in text; no "shortcut" in figure 1.

A2-1. We added "shortcut" in the Figure 1.

Change: Figure 1.

A2-2. Figure 1 is not needed to be cited here because Figure 1 is the actual network architecture used in this study and this part explains the characteristics of the DnCNN, thus we remove the sentence "The architecture of the DnCNN is shown in Fig. 1 and will be explained in more detail below". Figure 1 is now firstly cited in "Network architecture" section and we added ReLU in text.

Change: Line 140-142.

We modified the sentence from "Fig. 1 shows the DnCNN architecture used in this study, where Conv and BN indicate convolution and batch normalization, respectively." to "Fig. 1 shows the DnCNN architecture used in this study, where Conv, BN, and ReLU indicate convolution, batch normalization, and rectified linear units (Krizhevsky et al., 2012), respectively."

A2-3. The network architecture using shortcut is called as residual learning and we explained the residual learning as "Residual learning was first suggested by He et al. (2016) and it applied residual block which consisted of several convolution processes and a shortcut connection to the neural network to overcome the problem of machine learning when networks delve deeper." Therefore, "residual learning" is not needed to be shown in Figure 1.

Q3. Line 103. "by adding noise-free seismic data (y) and noise (n)" -> "as a sum of "true" seismic signal (y) and noise (n)"

A3. We modified the sentence

Change: Line 109-110.

- The seismic data including noise (y) can be expressed as a sum of true seismic data (x) and noise (n) as follows:

Q4. Lines 108-116. This section is difficult for oceanographers owing to the use of several unfamiliar and unexplained terms: residual learning, residual network, network depth (and why should it increase?), residual unit, trainable nonlinear reaction diffusion, batch, batch normalization (there is a gap to its explanation), mini-batch.

A4. We tried to explain many of the difficult terms and removed several unnecessary terms. However, some terms are so basic that even beginners who are not very interested in artificial intelligence already know them. Currently, many researchers in many fields are very interested in artificial intelligence, and so is the oceanographers. Therefore, some terms were not explained because they were unnecessary to explain in the paper.

Changes: Line 116-117, Line 119-120, Line 104-105, Line 124-125.

We modified the sentence from "it is different from the conventional residual network" to "it is different from the conventional neural network using residual learning (residual network)" to explain what is residual network.
We removed trainable nonlinear reaction diffusion which is unnecessary.

- We modified "residual unit" to "residual block". We modified the sentence from "Residual learning was first suggested by He et al. (2016) and it added the shortcut connection to the neural network to overcome the problem of machine learning when networks delve deeper." to "Residual learning was first suggested by He et al. (2016) and it applied residual block which consisted of several convolution processes and a shortcut connection to the neural network to overcome the problem of machine learning when networks delve deeper." to supplement the residual learning and explain what is residual block.

- We added the sentence "Instead of using entire training data at the same time, the machine learning sequentially uses mini-batch which is a small part of the entire data as input for the efficient training." to explain the minibatch. Since the explanation of mini-batch has been supplemented, the batch normalization can be better understood by readers.

Q5. Lines 133-140. More difficulty for oceanographers owing to unfamiliar and unexplained terms: rectified linear units (previously mentioned but not explained), activation function, why add nonlinearity? Is "3x3xc" and reference to colour (c=3) necessary since (line 140) c = 1? Why 64 feature maps?

A5-1. If there is not a nonlinear activation function, the network with many layers can be expressed single layer network because the result of Matrix1  $\times$  Matrix2  $\times$  ...  $\times$  MatrixN is the same as a single matrix Matrix0 (matrix multiplication is linear operation). However, if we add nonlinearity between the matrix multiplication, the multiplication cannot be expressed with a single matrix. This is why the nonlinear activation function is needed and more detail explanation can be found in Huang and Babri (1998).

Change: Line 146.

- We added the reference of Huang and Babri (1998).

Huang, G. B., and Babri, H. A.: Upper bounds on the number of hidden neurons in feedforward networks with arbitrary bounded nonlinear activation functions, IEEE transactions on neural networks, 9(1), 224–229, doi: 10.1109/72.655045, 1998.

A5-2. We modified and added the explanations of unfamiliar terms.

Changes: Line 146-150.

- We removed the sentence "The conventional DnCNN performs denoising from the image file (.jpg, .png, etc.), thus the size of the convolution filter is 3 × 3 × 3 in the color image and 3 × 3 × 1 in the gray image." because we did not use image file but binary file.
- We added "which is the same number of feature maps in Zhang et al. (2017)"
- We added the explanation of ReLU as "ReLU which returns input value for the positive input and 0 for the negative input is used for the activation function in this study"

Q6. Line 159. "multiplying by the square root of time"

A6. We modified "multiplying the square root of time" to "multiplying by the square root of time". Change: Line 170.

Q7. Line 166. "Internal waves in the research area propagate"

A7. We modified "internal wave of the research area propagates" to "internal waves in the research area propagate" Change: Line 179.

Q8. Line 172. "(assuming a constant density 1 g/cm3)" is only necessary for the XBT data. If you applied it to the XCTD data, did you ignore conductivity?

A8. We assumed a constant density for XBT and used measured density for XCTD.

Change: Line 185-186.

We modified the sentence from "calculated with the XBT and XCTD data (assuming a constant density 1 g/cm3)" to "calculated with the XBT (assuming a constant density 1 g cm-3) and XCTD data"

Q8. Line 189. "that" -> "those".A8. We modified "that" to "those"Change: Line 206.

Q9. Lines 225, 227-228. What do you mean by "epoch", "iteration"? "is a process" does not explain them. A9. We rewrote the explanation of "epoch" and "iteration."

Change: Line 242-249.

- The modified sentence is "In this study, training data were newly generated at every epoch with the fit\_generator function in Keras (Keras Documentation, 2020). Therefore, the number of training data used in the training is determined by the size of mini-batch and the number of iterations per epoch. The number of training using a mini-batch is called as iteration and the number of training using entire training data is called as epoch. If one mini-batch pass through the training, one iteration ends. If all mini-batches pass through the training and entire training data has been used for training, one epoch

ends. The mini-batch size was 128 and the number of iteration of an epoch was 220, thus the fit\_generator function generated 28,160 training data patches at every epoch."

Q10. Line 233. "or" -> "of" A10. We modified "or" to "of" Change: Line 256.

Q11. Line 234. What is the purpose of using the "Adam optimizer"?

A11. Adam optimizer minimizes the differences between the true data and estimated data during training. It gives better convergence than many other optimizers. Because the Zhang et al (2017) used adam optimizer in DnCNN, we also used adam optimizer. The adam optimizer is one of the most popular optimizers in the machine learning field and we gave the reference of the adam optimizer, thus we did not modify the sentence.

Q12. Line 332. "point" -> "patch". A12. We modified "point" to "patch" Change: Line 356.

Q13. Line 340. "data slope" and "horizontal slope" make no sense. You have not given the definition asked for by Referee 1.

A13. We modified the sentence by following Referee 1's suggestion

Change: Line 366-369.

- The sentence was modified to "The data slope spectrum is a slope spectrum obtained directly from seismic amplitude instead of tracked seismic reflections. The obtained horizontal wavenumber  $(k_x)$  spectrum of the seismic reflection amplitude is multiplied by  $(2\pi k_x)^2$  to produce a data slope spectrum, which is useful in identifying noise contamination of seismic data to reveal the cutoffs from an internal wave to turbulence subrange"

Q14. Lines 395-396. ". . scaled the seismic sections again, multiplying the signal by the square-root of return time at each time step to make the amplitude correction."

A14. We modified the sentence.

Change: Line 423-425.

- The sentence was modified to "we scaled the seismic sections again, multiplying the signal by the square root of time at each time step (consequently multiplying the time at each time step) for the spherical divergence correction."

Q15. Pages 40-42, 45 are blank A15. We removed blank pages.

**Referee 1 comments**

The authors have addressed my comments satisfactorily. The presentation is also improved with the revised figures. I think the paper can be published in OS.

We thank Referee 1 for detailed comments that improved the manuscript. Followings are the responses to each comment.

- A few minor technical issues:
- Q1. There are several blank pages.
- A1. We removed blank pages.

Q2. li340. I think the description of the data slope spectrum is still confusing for the reader. Perhaps the narrative could be: obtain the horizontal wavenumber spectrum of the seismic reflection amplitude; multiply this by  $(2\primes x_x)^2$  to produce a so-called "slope spectrum".

A1. We modified the sentence.

Change: Line 367-369.

- We modified the sentence as "The data slope spectrum is a slope spectrum obtained directly from seismic amplitude instead of tracked seismic reflections. The obtained horizontal wavenumber  $(k_x)$  spectrum of the seismic reflection amplitude is multiplied by  $(2\pi k_x)^2$  to produce a data slope spectrum, which is useful in identifying noise contamination of seismic data to reveal the cutoffs from an internal wave to turbulence subrange (Holbrook et al., 2013; Fontin et al., 2017)."

Q3. li 421, which generate data

A3. We modified "generates" to "generate". Change: Line 449.

Q4. Conclusions: perhaps rename as "Summary". A4. We modified "Conclusions" to "Summary" Change: Line 447.

Q5. li 437-438: the attenuation of the random noise is repeated in two sentences.

A5. We modified the sentence and rewrote the paragraph.

Change: Line 465-473.

- We removed "The observed random noise is successfully attenuated in the seismic section" and modified the sentence as "First, the calculated data slope spectrum indicates that a noise with a slope of  $k_x^2$  is not removed completely at wavenumbers above 0.02 cpm." The rest of the paragraph is rewroted as "Therefore, future studies should include a detailed analysis of the slope spectra of the SO data and establish an improved noise attenuation algorithm suitable for higher wavenumbers. Moreover, the data were collected and processed using 2D seismic exploration technology, which cannot efficiently deal with out-of-plane contamination. We expect that 3D seismic exploration can improve the resolution of SO data."

Q6. references: The authors do not give DOIs for the references.

A6. We added DOIs for the references where available.

Q7. conference papers: In physical oceanography papers, citation of conference papers, expanded abstracts are not very meaningful. We even avoid citing unpublished papers, these etc.

A7. We also notice that citing the conference papers may not be very meaningful in many of research area.

However, application of machine learning to the seismic data processing has become popular recently. Therefore, there are not many published full papers yet but many of studies are introduced in conference. Because of this reason, some of the references are citing the conference papers. When the expanded abstract was published in full paper, we modified the citation such as Liu et al. (2020).

Liu, D., Wang, W., Wang, X., Wang, C., Pei, J., and Chen, W.: Poststack Seismic Data Denoising Based on 3-D Convolutional Neural Network, IEEE Transactions on Geoscience and Remote Sensing, 58(3), 1598-1629, doi: 10.1109/TGRS.2019.2947149, 2020.

Q8. I was hoping the paper was going to be shorter but it came back with 20 figures. Font size in Figs 18, 19 and 20 can be larger. Sub-panel labels (a, b, c, d) can be moved to side or into the panels.

A8-1. This is the first study of applying the machine learning to attenuate the noise from the SO data. Therefore, the manuscript needed to include many detailed information of the machine learning and SO. In addition, we had to add a synthetic data test during the first revise, which required additional sub-section and several figures. For these reasons, the manuscript has been extended, but we think it is essential.

A8-2. We modified the fond size of Figs 18, 19, 20. The labels of the sub-panels can be relocated during the publication stage to make more suitable for publication.

**Referee 2 comments**

General comments– The paper has an improved balance of content, though I still think much of the detail of the filtering method could be moved to supplementary material as it will be of low interest to the principal readership of this journal.

Editor to advise if American spelling is acceptable and some phasing is still awkward in places but it does not detract from the understanding.

**We thank Referee 2 for detailed comments that improved the manuscript.**

Our study, we think, is the first attempt to attenuate noise in SO data using machine learning. Therefore, the readers will benefit much from our study if detailed procedures are illustrated in the manuscript. Followings are the responses to the comments and corrections.

Comments and corrections. Q1. line 8 punctuation A1. We modified "." to "," Change: Line 8.

Q2. line 23: "dropping equipment" ? Rephrase as this is ambiguous

A2. We rephrased this sentence.

Change: Line 23-25.

- The sentence was modified from "Conventional physical oceanography measurements from cruises are performed by dropping equipment at observation stations" to "Conventional physical oceanography measurements from cruises are performed by deploying instrument such as a conductivity/temperature/depth (CTD), an expendable conductivity/temperature/depth (XCTD) or an expendable bathythermograph (XBT) at observation stations."

Q3. line 25: first published use of technique was by Gonella, J., Michon, D., 1988. Deep internal waves measured by seismic-reflection within the eastern Atlantic water mass. Comptes Rendus de l Academie Des Sciences Serie II 306, 781–787 (in French with English abstract). However it was Holbrook who first used the term seismic oceanography.

A3. We added Gonella and Michon (1988).

Gonella, J., and Michon, D.: Ondes internes profondes rèvèlèes par sismique rèflexion au sein des masses d'eua en atlantique-est, C. R. Acad. Sci., Ser. II, 306, 781–787, 1988.

Change: Line 28-31.

- The added sentence is "Seismic oceanography (SO) is a method that obtains the water column reflections via seismic exploration and analyzes seismic sections to estimate the oceanographic characteristics of sea water. It was firstly attempted by Gonella and Michon (1988) to measure deep internal waves in the

eastern Atlantic and later became widely known after the work of Holbrook et al. (2003)."

Q4. Line 27/28: "The differences in temperature and salinity between water column generate ..." poor English A4. We modified the sentence.

Change: Line 31-34.

- The sentence was modified from "The differences in temperature and salinity between water column generate the difference in acoustic impedance, which reflect the seismic signals, and the reflected signals recorded at the receivers are processed to image the thermohaline fine structure of the ocean" to "The difference in temperature and salinity between water mass generates an acoustic impedance contrast, resulting in the reflection of seismic signals. The reflected seismic signals are processed to image the thermohaline fine structure of the ocean (Ruddick et al., 2009)"

Q5. line 29: need reference here suggest Ruddick, B., Song, H., Dong, C., Pinheiro, L., 2009. Water column seismic images as maps of temperature gradient. Oceanography 22 (1), 192–205.

A5. We added a reference.

Change: Line 34.

Q6. line 31: need reference for "conventional oceanographic methods" and link to the equipment that was dropped in line 23.

A6. We modified the sentence and added a reference.

Change: Line 36.

We modified the sentence from "it has the advantage of generating data with a high horizontal resolution compared to conventional oceanographic methods" to "it has the advantage of generating data with improved horizontal resolution over conventional probe-based oceanographic methods (Dagnino et al., 2016)."

Q7. 34: redundant word "careful"

A7. We removed "careful"

Change: Line 40.

**Q8. line 39: "...higher exploration expenses..." ambiguous as higher than what**

A8. We modified the "higher exploration expenses" and added explanation.

Change: Line 45-46.

- We modified sentence from "SO also has the disadvantage of higher exploration expenses when using air guns and streamers that are several kilometers long" to "SO also has the disadvantage of high exploration expenses when using air guns and streamers of several kilometers long, which require large vessel and many operators."

Q9. line 45: the vertical resolution of 1.5 m is based on the theoretical Rayleigh limit however in a 3D environment it is unlikely that you can achieve this due to interference from out-of-plane scattering.

A9. Piété et al. (2013) indicated that their data obtained by using SIG sparker with a 250Hz central frequency has the vertical resolution of 1.5 m, thus we mentioned "a high vertical resolution of 1.5m". They also seem to ignore the out-of-plane scattering issue, thus we modified the sentence

Change: Line 51.

- The sentence was modified from "a high vertical resolution of 1.5m" to "a high vertical resolution of approximately 1.5m"

Q10. Line 49/50: the maximum impedance contrasts are smaller

A10. We modified "impedance contrasts" to "maximum impedance contrasts." Change: Line 55-56.

Q11. line 53 & 64: most noise sources are not random so the authors need to define which noise sources that consider as problematical, especially for SO data where a dominant noise is caused by pressure variations at the receiver due to waves on the surface. This peaks at frequencies below ~5 Hz so with sparker SO you should do better than airguns as you can set your low-cut filter at a higher frequency without damaging your primary signal bandwidth. I think what the authors are looking at here is noise generated from the ship as it has a distinct time characteristic and I would not be surprised if this was during periods of tidal flow against the direction of travel. So perhaps the ships propellers were run at a faster rate to maintain speed over the ground which has resulted in more cavitation. However, that period does not match with the statement that the ship speed was 5.5 knots. Suggest authors include auto-correlation functions both in space and time of the red boxes (Fig 3) indicated to justify that the synthetic random noise used in training datasets is appropriate.

A11. In seismic exploration, the term "noise" refers to the unwanted responses which are not meet the purpose of the exploration. There are many sources of noise such as feature of seismic source, ship, wave, weather, and even marine organisms (plankton, fish, etc.). The noise which has coherent characteristic is called as "coherent noise" (multiple, bubble effect, backscattering, etc.) and this can be removed during the seismic processing stage. The noise excluding coherent noise is generally referred to as "random noise". Directly removing random noise is difficult and random noise usually attenuated during the stacking procedure which can increase S/N ratio. However, when the random noise is strong and does not present white characteristics in the frequency band, removal becomes very difficult.

For the conventional noise attenuation methods, it was very important to estimate and analyze the source of noise and apply the appropriate algorithm. On the other hands, the machine learning proposed in this study uses random noise already measured in the survey area, thus the characteristics of noise for various sources can be automatically considered. Even the noise generated by the source that we have not yet figured out can also be taken into account. Of course, estimating and analyzing the source of noise is a very important issue, but we think detailed analysis of the actual source of the noise is a little off from the machine learning perspective which is the main content of the paper. We added the explain of the possible source of the random noise in the seismic data and advantage of using suggested machine learning in the "East Sea SO data" section.

Change: Line 190-193.

We added the sentence "Random noise in the seismic data can be created by rough weather conditions, ocean swells, a tail buoy jerk, the engine and the propeller of the vessel, etc. For the conventional noise attenuation methods, it is important to estimate the noise sources and their properties. In contrast, for machine learning-based noise attenuation suggested in this study, estimation of those propertied is not needed because noise itself is used as training data."

Q12. Figure 2 – does this show the whole line or just the portions after the shelf sections were omitted?

A12. The solid line in Figure 2 was the whole line of the exploration. We modified the Figure 2 to show the whole survey line and the study area simultaneously.

Change: Figure 2.

- The caption of Figure 2 was changed from "The black solid line is the survey line, and the black dashed lines with arrow indicate the exploration directions of lines 1 and 2, and red dots are the locations of XBTs and XCTDs." to "The solid line with gray and black color is the survey line. Gray line is shelf and slope parts which were removed from the seismic section during data processing and black line is the target area of this study. The black dashed lines with arrow indicate the exploration directions of lines 1 and 2, and red dots are the locations of XBTs and XCTDs."

Q13. Line 163: no comma in velocity value and possibly need to correct style to m s-1 and other units as necessary. Also this is ambiguous as 'water velocity' in the context of oceanography describes the mass movement of a body of water. An acceptable term is 'sound-speed'. As the temperature and salinity of the water is known (XCTD) then authors could do better than just use a generic sound speed by using the equations of state.

A13. We modified "constant velocity of 1,500 m/s" to "constant sound speed of 1500 m s-1". We also modified "g/cm3" to "g cm-3". It would be better using true temperature and salinity of the water to calculate the sound speed of water instead of using 1500m/s. However, the difference between true sound speed and 1500 m/s is small and does not give meaningful differences in the seismic imaging. Because of this reason, several studies such as Tsuji et al. (2005), Holbrook et al. (2013), piete et al. (2013), Moon et al. (2016) also used constant sound speed for seismic data processing.

Changes: Line 174-175.

Q14. Line 173; define what you mean by very small as it looks like the calculated reflection coefficients are at the level of the minimum resolution of the XBT and given that you have two XCTD why did you use a constant density for those?

A14. We added reflection coefficient value in the sentence. We calculated reflection coefficients of the XCTD

data by using measured sound speed and density and rewrote the sentence.

Changes: Line 185-186.

- The sentence was modified from "The reflection coefficients are very small" to "The reflection coefficients are very small (~0.00005)". Figure 2 was re-plotted.
- The sentence "calculated with the XBT and XCTD data (assuming a constant density 1 g cm-3)." was modified to "calculated with the XBT (assuming a constant density 1 g cm-3) and XCTD data."

Q15. Line 209: justify why you used 1000 kg m-3 for the density as true density is provided as part of the model – why does this change make the synthetic more SO-like if you don't apply a scaling an shift to the compressional wave velocities too. It sounds like a fix to overcome inefficiencies in your modelling process eg calling depth time and followed by a convolution. Given the level of adjustment and modification (see line 224), I see no purpose for using these sophisticated models as a set of time-shifted spikes (eg an enlarged version of Fig 16) would have probably done equally as well.

A15. To generate the noise-free synthetic seismic section, appropriate synthetic impedance model is necessary. There are several ways to obtain the synthetic impedance model: generating the synthetic model manually, modifying the existing synthetic model, directly using the existing synthetic model, etc. If we want to generate the synthetic model manually, it needs additional programing and computing cost. The easiest way is modifying existing synthetic model. The adjustment and modification of the synthetic models performed in this study is much easier and requires less time and effort compared to manually generating the synthetic model (similar to the model in Figure 16). Therefore, we used Marmousi and Sigsbee2A synthetic model to generate the synthetic seismic section.

There is no advantage of using synthetic true density instead of constant density in the generation of synthetic seismic section. Therefore, we just assumed constant density.

Q16. Line 210: were the simulations acoustic or elastic?

A16. Since we used acoustic impedance model, it is acoustic.

Q17. Line 247-260: you don't actually know the 'ground truth' as the section without added noise is, as you correctly point out, still contaminated with noise and will contain inter-layer peg-legs and out-of-plane events which you algorithm will also treat as primary signal.

A17. The SEZ field data still contains the noise because it is almost impossible to remove all kinds of noise in the seismic data. However, the "ground truth" term of the machine learning is the "true value" in the misfit function of the machine learning and the SEZ field data is used as the "true value" in the training data 1, thus we used "ground truth".

Q18. Line 275: what is x and y in equation 5 A18. We explained what is x and y. Change: Line 299.

- The sentence is modified from " $\sigma_{xy}$  is the covariance of x and y" to " $\sigma_{xy}$  is the covariance of reference image (x) and test image (y)."

Q19. line 339 suggest a sub heading here to indicate you have changed to dealing with the SO data A19. We added subheading "3.4 Calculation of the data slope spectrum from the synthetic seismic section" Change: Line 364.

Q20. line 342 "and"  $\rightarrow$  "to reveal" A20. We modified "and" to "to reveal" Change: Line 369.

Q21. lines 342-344 suggest: Holbrook et al. (2013) suggested analysing the slope of the spectrum for the complete data before calculating the slope of the spectrum from the water reflections because the noise that needs to be suppressed is more evident in the spectrum of the complete data.

A21. We modified the sentence

Change: Line 370-372.

- The modified sentence is "Holbrook et al. (2013) suggested analyzing the slope of the spectrum for the complete data before calculating the slope of the spectrum from the water reflections because the noise that needs to be suppressed is more evident in the spectrum of the complete data."

Q22. line 369 "clearly imaged" → "improved"
A22. We modified "clearly imaged" to "improved".
Change: Line 397.

Q23. Line 375-378, 385-387 & 389-393: I think you are making too much of this as the differences between the approaches is marginal as shown by your fig 20. Essentially both methods worked on the data to a sufficient degree to enable meaningful estimates of the slope of the internal wave to be discerned. If you feel you want to emphasise the benefit of D2 vs D1 then show a difference plot of the two.

A24. We removed comment of several minor differences between D1 model result and D2 model result. Change: Line 404-406, Line 414

Q24. Line 397: what sound speed model did you use for conversion?

A24. We used constant sound speed model of 1500 m/s.

Change: Line 425.

- We added "using a constant sound speed of 1500 m s-1"

Q25. Line 406: would be useful to quote horizontal limit of resolution due to the Fresnel zone which I estimate, by eye-balling your data, to be  $\sim$ 0.02 cpm so above this value you would expect the amplitude to the perturbations to be strongly filtered so the fact the signal becomes noise limited is to be expected and further noise reduction will not provide any useful information. I note that your data appear to have a lower frequency content than I would expect – how do you explain that? What were the acquisition parameters? As these directly effect the data signal bandwidth.

A25. The horizontal resolution due to the first Fresnel zone radius (r) is approximately 17.3 m when the dominant frequency is 250 Hz, depth is 100 m (middle depth of the target zone), and sound speed is 1500 m/s based on following equation (Yilmaz, 2001)

$$r = \sqrt{\frac{z_0\lambda}{2}}$$

 $z_0$  is depth of reflector and  $\lambda$  is wavelength. The horizontal resolution of the depths ranging from 30 m to 200 m is approximately 9.5 m to 24.5 m.

The receiver interval is 6.25 m and we made super gather by using 4 neighboring CMPs (resulting 12.5m (0.08 cpm) interval of super gather CMP). Therefore, the horizontal resolution of the seismic data is larger than CMP interval at the section deeper than 52.1 m (r=12.5 m when depth is 52.1 m).

The horizontal wavenumbers of 30 m depth and 200 m depth is approximately 0.1 and 0.04 cpm, respectively. It means that if we can remove the noise at wavenumbers above 0.02 cpm, it would be possible to obtain more information. We mentioned that we made super gather during the processing stage and added the vertical and horizontal resolution of the East Sea data.

Change: Line 177-179.

We added "The calculated vertical and horizontal resolution (Yilmaz, 2001) of the processed seismic section are approximately 1.5 m and 17.3 m when the central frequency is 250 Hz, sound speed is 1500 m s-1, and depth is 100 m." to give the information of vertical and horizontal resolution in the "2.3 East Sea SO data" section.

[revised manuscript text omitted]

---

## Author Response (AR3)

Thank you for your careful review and constructive comments. We have studied all of your comments carefully and revised our manuscript. Followings are the response to the Editor. "Q" is a comment from the reviewer, "A" is a response to the comment and "Change" the lines and details of the modification. The comments are black in color and responses / changes are blue in color.

**Topic Editor Decision: Publish subject to minor revisions (review by editor)** (03 Sep 2020) by John M. Huthnance
Comments to the Author:
Dear Authors

Thank-you for your re-revised manuscript with responses to the referees. I am now asking for minor revisions, as below. This means that I (but not reviewers) should see it again: the main reason is to check that it is intelligible to oceanographers. You argue reasonably that the machine learning aspects should be presented, but there is little value in that if readers cannot understand the words or "jargon" used. Please make every attempt to use standard English and explain technical terms.

→ We tried to explain difficult technical terms, and removed several unnecessary sentences to make manuscript easier to understand. We also modified some phrases for standard English.

**Detailed comments.**

Q1. Line 30. Better ". . (2003). Spatial differences in temperature and salinity generate . ."

A1. We modified the phrase.

Change: Line 30.

- From "The difference in temperature and salinity between water mass generate" to "Spatial differences in temperature and salinity generates…"

Q2. Lines 42-43. Better ". . streamers several kilometres long, . ."

A2. We modified the phrase.

Change: Line 43-44.

- From "… streamers of several kilometers long…" to "…streamers several kilometers long…"

Q3. Line 47-48. Given your response to Referee 2 Q9, maybe better ". . data with vertical resolution theoretically as fine as 1.5 m were acquired . ."

A3. We modified the phrase.

Change: Line 49.

- From "data with a high vertical resolution of approximately 1.5 m were acquired" to "data with vertical resolution theoretically as fine as 1.5 m were acquired"

Q4. Lines 100-102. This sentence is not clear. What does "it" refer to? What is a "residual block"? Either "residual blocks" or "a residual block". Surely machine learning is wanted and not a "problem" "to overcome"?

A4. The detailed explanation of the residual learning and the problem related to the deep neural network is unnecessary for this paper and will make paper more difficult, therefore we modified the sentence. We added the explanation of the residual block. We also modified the sentence explaining "residual network" which were in line 113 because now the "residual network" is explained in line 101-102.

Changes: Line 101-104.

- From "Residual learning was first suggested by He et al. (2016) and it applied residual block which consisted of several convolution processes and a shortcut connection to the neural network to overcome the problem of machine learning when networks delve deeper." to "Residual learning was first suggested by He et al. (2016) and the neural network which uses residual learning (residual network) contains several residual blocks. The residual block is a building block of the residual network and consists of several convolution processes and a shortcut connection."

Changes: Line 114-115.

- From "it is different from the conventional neural network using residual learning (residual network)" to "it is different from the conventional residual network."

Q5. Lines 236-238. These two sentences might better be at line 234 before first use of "epoch". If I understand the sentence at lines 236-237, then better "Each training cycle using one mini-batch is an "iteration" and each training cycle using all the training data is an "epoch"."

A5. We modified the sentences and moved four sentences related to the explanation of "iteration" and "epoch" to line 234.

Change: Line 237-245.

- The modified sentences are "The number of training data used in the training is determined by the size of the mini-batch and the number of iterations per epoch. Each training cycle using one mini-batch is an "iteration" and each training cycle using all the training data is an "epoch". If one mini-batch passes through the training, then one iteration ends. If all mini-batches pass through the training and all the training data has been used for training, then one epoch ends."

Q6. Lines 256-258. "Eighty-six $300 \times 300$ size test data were available, and we divided the test data into $50 \times 50$ size patches, which is the same size as the training data patch. Then, we discarded the remaining data divided by 128 (mini-batch size) for computational efficiency; thus, the number of test data points was 3,072." I do not follow this arithmetic. There can be 86 x 36 = 3096 patches in your test data. What exactly was discarded and what was divided by 128? Do you mean "points" or "patches"?

A6. We agree you that the arithmetic should be clear. We clarified the arithmetic and made the complex sentence clear.

Change: Line 262-265.

- From "Eighty-six $300 \times 300$ size test data were available, and we divided the test data into $50 \times 50$ size patches, which is the same size as the training data patch. Then, we discarded the remaining data divided by 128 (mini-batch size) for computational efficiency; thus, the number of test data points was 3,072." to "Eighty-six $300 \times 300$ size test data were divided into $50 \times 50$ size patches, which is the same size as the training data patch, and 3,096 patches $(50 \times 50)$ were generated. Among 3,096 patches, we used 3,072 patches which was 24 times of the mini-batch size (128) for the test because of computational efficiency."

Q7. Line 293 and figures 9, 13. Does "before" mean before the first epoch? If so, please make this clear in the text, the figures and especially the figure captions. At present "before" might mean after the 39th epoch, indeed that is what the figure captions definitely imply.

A7. We modified "before" to "before applying the D1 (or D2) model" to make sentence clear.

Changes: Line 301 and 339, Figure 9 and 13.

Q8. Line 311. "points" –> "patches"?

A8. We modified "points" to "patches".

Change: Line 319.

Q9. Line 409. "multiplying the time" –> "multiplying the spectrum by time"? It would help if you made clear that "time" here is a proxy for reflector distance.

A9. We modified the phrase to make clear.

Change: Line 417.

- From "consequently multiplying the time at each time step" to "consequently multiplying the seismic signal by the time at each time step"

Q10. Line 437. "which" –> "and". [What you have said here is that an air gun source can provide information with higher vertical resolution. Only use "which" immediately after the item it refers to.]

A10. We modified "which" to "and".

Change: Line 445.

Q11. Data Availability. You should put the data in a long-term accessible data bank that does not depend on the authors.

A11. We uploaded our DnCNN python code, Marmousi2 synthetic data and field noise data on Zenodo (https://www.doi.org/10.5281/zenodo.4020335) by following Ocean Science data and code availability instruction.

Change: in "Data availability" section

- The code, synthetic training data samples, field noise data are available at https://www.doi.org/10.5281/zenodo.4020335. Marmousi 2 model is available at https://wiki.seg.org/wiki/AGL_Elastic_Marmousi, Sigsbee2A model is available at http://www.delphi.tudelft.nl/SMAART/, and 1994 BP statics benchmark model is available at https://wiki.seg.org/wiki/1994_BP_statics_benchmark_model. The East Sea sparker field data can be made available upon request to authors.

Q12. Need to add Gonella and Michon (1988).

A12. We added Gonella and Michon (1988), and modified other errors in "Reference"

Changes: in "Reference" section,

- we modified "doi:" to "https://doi.org/"

[revised manuscript text omitted]